# ML-Agent: Reinforcing LLM Agents for Autonomous Machine Learning Engineering

## Abstract

The emergence of large language model (LLM)-based agents has significantly advanced the development of autonomous machine learning (ML) engineering. However, the dominant prompt-based paradigm exhibits limitations: smaller models lack the capacity to learn from execution trajectories for generalization, while large proprietary models incur high computational overhead, restricting accessibility and scalability. Focusing on this, for the first time, we explore the paradigm of learning-based agentic ML, where an LLM agent learns through interactive experimentation on ML tasks using online reinforcement learning (RL). To realize this, we propose a novel agentic ML training framework with three key components: (1) exploration-enriched fine-tuning, which enables LLM agents to generate diverse actions for enhanced RL exploration; (2) step-wise RL, which enables training on a single action step, accelerating experience collection and improving training efficiency; (3) an agentic ML-specific reward module, which unifies varied ML feedback signals into consistent rewards for RL optimization. Leveraging this framework, we train ML-Agent, driven by a 7B-sized Qwen-2.5 LLM for autonomous ML. Despite training on only 9 ML tasks, our 7B-sized ML-Agent achieves comparable performance to agents using much larger proprietary LLMs (e.g., GPT-5) but at significantly lower computational cost, demonstrating strong performance and cross-task generalization.

## 1 Introduction

Machine Learning (ML) engineering is a critical yet labor-intensive process, requiring expert researchers to invest significant time—potentially days or even months—designing architectures, tuning parameters, and iteratively refining models through trial and error (Bergstra & Bengio, 2012). This challenge has sparked an ambitious vision of autonomous ML: building autonomous AI systems that independently orchestrate the entire ML lifecycle, from conceptual design and code implementation to refinement.

Fortunately, the advent of LLM-based agents, equipped with capabilities of interaction (Du et al., 2023; Pang et al., 2024), coding (Hong et al., 2023; Qian et al., 2023; Hu et al., 2024) and tool-calling (Masterman et al., 2024), has propelled us significantly closer to realizing this vision (Agentic ML) (Huang et al., 2023; Chan et al., 2024). Unlike traditional automated ML with pre-defined limited search and action spaces (Tang et al., 2024; LeDell & Poirier, 2020; Jin et al., 2023), these LLM agents, when provided with instructions in natural language, can autonomously propose effective actions, generate executable codes, and iteratively improve solutions based on environmental feedback (Huang et al., 2023; Jiang et al., 2025). For example, AIDE (Jiang et al., 2025) and ML-Master (Liu et al., 2025a) both leverage LLM agents together with experimental environments to automate ML process.

Currently, the dominant paradigm in agentic ML relies on prompt-based design, where agents are constructed through heuristic prompt engineering. This approach offers practical advantages, as it allows rapid deployment without parameter updates or extensive retraining. However, it also exhibits notable limitations: when driven with smaller language models, such agents lack the capacity to learn from and internalize execution trajectories, causing limited generalization across diverse tasks; conversely, when implemented with large-scale proprietary models, the paradigm incurs substantial computational overhead and resource consumption, thereby restricting accessibility and undermining sustainable scalability (Belcak et al., 2025).

To address these limitations, we propose moving beyond the prompt-based paradigm toward a new research trajectory: learning-based agentic ML. In this paradigm, agents are no longer constrained to static prompt instructions but instead learn adaptively from task-solving trajectories via online reinforcement learning (RL). Such a formulation empowers agents to systematically explore diverse strategies, accumulate knowledge across successive runs, and progressively refine their decision-making processes (Xiong et al., 2024). Importantly, this learning-based approach endows even relatively small language models with the capacity to achieve strong generalization, while substantially reducing computational and resource demands. As a result, it opens a more accessible, efficient, and sustainable path for advancing the frontier of autonomous machine learning.

While being straightforward, employing online RL to train autonomous ML agents poses three key challenges. (1) *Limited exploration*: agents often propose similar actions for the same ML task across runs, leading to narrow exploration trajectories in RL (Park et al., 2024). (2) *Slow experience collection*: ML experiments can take minutes to hours, making online RL data gathering inefficient and thus limiting feedback-driven training samples (Chan et al., 2024). (3) *Complex reward design*: agentic ML involves various outcomes, such as task-specific metrics, out-of-memory failures, and compilation errors. This requires a unified reward function to reconcile varied feedback signals (Eschmann, 2021).

In response to these challenges, we propose a novel agentic ML training framework, the first designed to train LLM agents for autonomous ML engineering using online RL. This framework enables agents to explore diverse ML trajectories, collect rewards efficiently, and iteratively enhance their capabilities through learned experience. (1) To improve exploration diversity, we introduce *exploration-enriched fine-tuning*, generating a diverse action pool from fast-executable ML tasks to finetune agents for broader RL exploration. (2) To accelerate experience collection, we design a *step-wise RL paradigm*, evaluating atomic actions using expert trajectories as single-step queries, significantly boosting training efficiency. (3) To tackle reward design, we develop an *agentic ML-specific reward module* that dynamically handles errors (e.g., runtime failures) and quantifies performance via normalized, task-specific metrics (e.g., accuracy gains).

By leveraging our proposed agentic ML training framework, we train ML-Agent, an agent driven by a 7B-sized Qwen2.5 LLM for autonomous ML. During training, our ML-Agent can efficiently explore the environment, learn from experience, and achieve continuous performance improvement through iterative exploration across various ML tasks. Surprisingly, despite its modest size and training on only **9** ML tasks, ML-Agent demonstrates strong performance and cross-task generalization, outperforming 671B-sized DeepSeek-R1 agent on 3 held-in and 10 held-out tasks across diverse data modalities and objectives. Notably, it achieves results comparable to agents using the most advanced proprietary LLMs (GPT-5) but at significantly lower computational cost.

In summary, our work makes the following significant contributions to the field:

- We introduce a new paradigm for autonomous ML: learning-based agentic ML, where an LLM agent learns through interactive experimentation on ML tasks via online reinforcement learning.
- We propose a novel training framework for agentic ML, which incorporates three technical designs: exploration-enriched fine-tuning, step-wise RL, and agentic ML-specific rewards.
- Extensive experiments show that despite training on only 9 ML tasks, our 7B-sized ML-Agent surpasses agents driven by much larger LLMs and even matches agents driven by proprietary LLMs (e.g., GPT-5) with much lower cost.

## 2 RELATED WORK

**Autonomous Machine Learning.** Autonomous machine learning aims to automate the manual and expertise-intensive aspects of machine learning, including data preprocessing, model selection and hyperparameter tuning. Autonomous machine learning has evolved from classical hyperparameter and pipeline search to agentic frameworks powered by large language models. Classical autonomous machine learning frameworks focus on automating model selection, hyperparameter optimization, and pipeline construction within a fixed search space (Tang et al., 2024; Olson & Moore, 2016; Feurer et al., 2022; Mohr et al., 2018; Erickson et al., 2020; Liu et al., 2020). For example, AutoGluon-Tabular (Erickson et al., 2020) ensembles multiple models and stackings to deliver state-of-the-art performance on tabular data with minimal user effort. These classical autonomous machine

learning works remain constrained by predefined search spaces and static configurations, lacking the adaptability and continuous learning capabilities.

**LLM Agents in Autonomous Machine Learning.** Recent advancements in LLMs have empowered them to autonomously generate and refine machine learning solutions, opening new possibilities in machine learning. Methods such as AutoML-GPT (Zhang et al., 2023b) and MLCopilot (Zhang et al., 2023a) prompt LLMs to automate the entire machine learning pipeline, where MLCopilot introduces past experience retrieval to help decision-making. AIDE (Jiang et al., 2025) and ML-Master (Liu et al., 2025a) focus on optimizing the ML engineering process through iterative search and refinement strategies. Other works like AutoKaggle (Li et al., 2024) and AutoML-Agent (Trirat et al., 2024) employ a multi-agent framework to address ML problems. However, these approaches are fundamentally constrained by a prompt-based paradigm. While agents may leverage past experience, their underlying models are not trained on these interaction histories. Consequently, their problem-solving strategies remain static and rely on costly advanced models. This limitation motivates our shift toward a learning-based paradigm where agents adapt and improve over time.

**Reinforcement Learning for LLMs.** Reinforcement learning (RL) significantly enhances the ability of LLMs, particularly in preference alignment and complex reasoning (Xu et al., 2025; Wang et al., 2024; Zheng et al., 2023). By facilitating exploration and exploitation, RL trains LLMs to adapt and improve their policy based on feedback, thus refining their performance in dynamic environments. One line of work is preference optimization (Kaufmann et al., 2023), with methods such as Reinforcement Learning from Human Feedback (RLHF)(Ouyang et al., 2022). RL is also utilized to train LLMs for complex reasoning tasks (Guo et al., 2025a; Liu et al., 2025b). Another line of research involves training LLM agents for specific tasks using RL (Zhang et al., 2025). For example, IPR (Xiong et al., 2024) and AgentQ (Putta et al., 2024) use DPO (Rafailov et al., 2023) to iteratively refine their policy. While StarPO (Wang et al., 2025) discusses the multi-turn reinforcement learning considering episode-wise reward. However, applying RL to train LLM agents for autonomous machine learning remains unexplored.

## 3 PROBLEM SETUP AND PRELIMINARIES

**Problem Formulation.** Agentic ML leverages an LLM agent to autonomously orchestrate the ML lifecycle by interacting with the experimental environment. This environment includes editable task-related code files together with an interpreter executing code and provides explicit experimental feedback (e.g., code execution results or error messages). Given an initial ML task specification (e.g., dataset description and evaluation metric), the agent begins interacting with the environment to iteratively refine its solution. At each step, the agent takes actions (e.g., add BN layers in the model architecture) and receives feedback (e.g., code execution output or error messages) from the environment. This loop continues until a step or time limit is reached. We follow the action space from prior work (Huang et al., 2023) (The details are provided in Table 4).

**Agentic ML as a MDP.** We format agentic ML as a Markov Decision Process (MDP) $\mathcal{M} = (\mathcal{S}, \mathcal{A}, \mathcal{P})$, where $\mathcal{S}$ is the state space, $\mathcal{A}$ the action space and $\mathcal{P}$ the state transition dynamics. Let the environment feedback at time $t$ be $f_t \in \mathcal{F}$, where $\mathcal{F}$ denotes the feedback space. We employ a history-based state representation $s_t = (s_0, a_0, f_0, a_1, f_1, \ldots, a_{t-1}, f_{t-1})$ to capture richer contextual information from past feedback, in which $s_0$ encodes the initial ML task specification and each pair $(a_i, f_i)$ represents the agent's action and corresponding environment feedback. The agent policy $\pi_\theta$ generates an action $a_t \in \mathcal{A}$ conditioned on current state $s_t$, forming a trajectory of interactions $\tau = (s_0, a_0, s_1, \ldots, a_{n-1}, s_n)$. Note that $\theta$ is the LLM's parameters within the agent and $n$ is the trajectory length. The goal is to maximize the expected trajectory reward:

$$\mathcal{J}(\theta) = \mathbb{E}_{\tau \sim \pi_\theta} \left[ R(\tau) \right], \tag{1}$$

where the reward function $R(\tau)$ denotes the cumulative reward over the entire trajectory.

**Challenges.** Although the formulation of agentic ML is relatively straightforward, employing online RL to train LLM agents for autonomous machine learning poses several key challenges, including: **(1) Limited exploration.** Agents often repeat similar actions across episodes, narrowing their exploration and limiting their ability to discover innovative ML solutions. **(2) Slow experience collection.** ML experiments can take minutes to hours, slowing down the online data collection process for RL training. **(3) Complex reward design.** Agentic ML produces varied outcomes (e.g., execution results

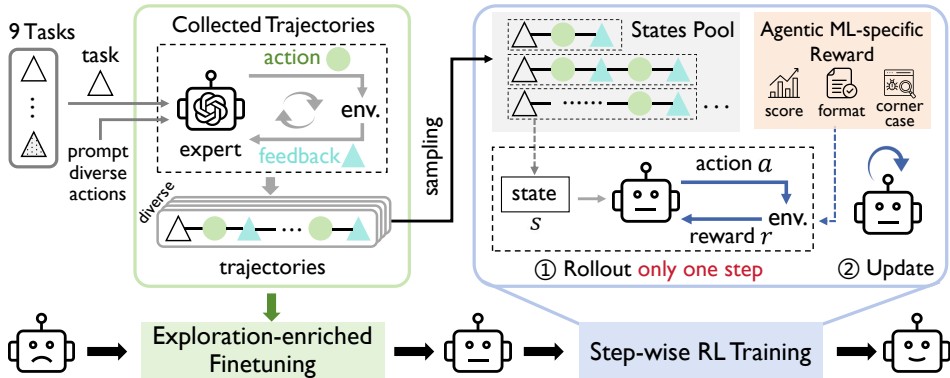

Figure 1: Overview of the agentic ML training framework, introducing (1) exploration-enriched fine-tuning for diverse action pool creation, (2) a step-wise RL paradigm for efficient experience collection using expert trajectories, and (3) an agentic ML-specific reward module for various ML feedback handling and task-specific performance evaluation.

or resource errors), making it challenging to design a unified reward function that effectively guides the agent. The subsequent section presents our agentic ML training framework designed to overcome these challenges, with the overall architecture illustrated in Figure 1.

## 4 AGENTIC ML TRAINING FRAMEWORK

Our agentic ML training framework is designed to train LLM agents for autonomous machine learning. As shown in Figure 1, it comprises three key steps for effective learning. First, *exploration-enriched fine-tuning* builds a diverse action pool to enhance RL exploration. Second, a *step-wise RL paradigm* uses expert trajectories as single-step queries to accelerate experience collection in RL. Third, an *agentic ML-specific reward module* handles errors and quantifies agentic ML task-specific performance. These steps sequentially enable diverse exploration, efficient training, and unified feedback, enabling agents to iteratively improve agentic ML performance across varied ML tasks.

### 4.1 EXPLORATION-ENRICHED FINE-TUNING

In agentic ML, limited exploration hinders autonomous machine learning workflows. Agents often repeat similar actions (e.g., small code edits) across episodes, leading to narrow exploration and preventing the discovery of innovative architectures or optimization strategies.

To address this, we introduce exploration-enriched fine-tuning with an automated data collection pipeline. It organizes ML optimization strategies into 3 semantic categories: data, model and learning. For each category, an LLM generates a large set of candidate ideas and an embedding-based diversity filter selects a compact and diverse pool. During trajectory generation, the system automatically samples 1–3 categories, shuffles their order, and draws one idea from each corresponding pool to form the initial action sequence (See Appendix B.1). An expert LLM with policy $\pi_e$ then executes the full workflow on fast-executable ML tasks, producing expert trajectories $\mathcal{D} = \{\tau^{(i)}\}_{i=1}^{|\mathcal{D}|}$. We fine-tune the agent policy $\pi_\theta$ via supervised fine-tuning (SFT):

$$\mathcal{L}_{\text{SFT}}(\theta) = -\mathbb{E}_{\tau \sim \mathcal{D}} \left[\log P_{\pi_\theta}(\tau|s_0)\right] = -\mathbb{E}_{\tau \sim \mathcal{D}} \left[\log \prod_{t=0}^{n-1} \pi_\theta(a_t|s_t)\right] = -\mathbb{E}_{\tau \sim \mathcal{D}} \left[\sum_{t=0}^{n-1} \log \pi_\theta(a_t|s_t)\right]. \quad (2)$$

This exploration-enriched fine-tuning approach preserves action format compliance while enabling agents to learn diverse strategies, significantly broadening the exploration scope in subsequent RL.

### 4.2 STEP-WISE RL PARADIGM

**Objective.** Due to the time-consuming nature of AI experiments, directly applying RL methods (e.g., PPO) is impractical, as sampling a single trajectory during rollout takes hours. To address

this issue, we propose a step-wise RL approach that reformulates the objective function equation 1, where we sample only a single step of action during the rollout phase instead of the entire trajectory. This approach extensively reduces the computational cost of the rollout phase and makes the overall training process more efficient. Specifically, we expand equation 1 into steps according to the state distribution $d^{\pi_\theta}(s)$:

$$\mathcal{J}(\theta) = \sum_{t=0}^{n-1} \sum_{s_t \in \mathcal{S}} d^{\pi_\theta}(s_t) \left[ \sum_{a_t \in \mathcal{A}} \pi_\theta\left(a_t | s_t\right) R(s_t, a_t) \right], \tag{3}$$

where $R(s_t, a_t)$ is the step-wise reward at time $t$, and $d^{\pi_\theta}(\cdot)$ is the state distribution at time $t$ under policy $\pi_\theta$. This distribution can be calculated recursively based on the policy $\pi_\theta$ and the state transition dynamics $\mathcal{P}$; see Appendix A for details. The time-consuming components in equation 3 include: 1) $d^{\pi_\theta}(s_t)$, which involves multiple state transition dynamics from $s_t$ to $s_{t+1}$, and 2) $R(s_t, a_t)$, where the reward is determined based on feedback from the environment (e.g., code execution platform). Since $d^{\pi_\theta}(\cdot)$ relies on $\pi_\theta$, the trajectory sampling process operates repeatedly in standard RL training, making the computational cost even higher. However, using $d^{\pi_\theta}$ to sample state distribution is not necessary for two reasons: 1) $\pi_\theta$ poorly aligns with the environment format during the early stage of RL training, hindering effective state exploration; 2) Once $\pi_\theta$ can interact properly with the environment, the set of states it could explore tends to vary only slightly as $\pi_\theta$ updates. Hence, we sample the states from a states pool according to a fixed expert distribution $d^{\pi_e}(s_t)$, which forms the step-wise objective function

$$\mathcal{J}_{\text{step}}(\theta) = \sum_{s_t \in \mathcal{S}} d^{\pi_e}(s_t) \left[ \sum_{a_t \in \mathcal{A}} \pi_\theta\left(a_t | s_t\right) R(s_t, a_t) \right] = \mathbb{E}_{s_t \sim d^{\pi_e}, a_t \sim \pi_\theta(\cdot | s_t)} \left[ R(s_t, a_t) \right]. \tag{4}$$

This objective function $\mathcal{J}_{step}(\theta)$ reformulates multi-step trajectory RL into step-wise training. This reformulation offers two advantages: 1) The state sampling process is decoupled from the RL of the model. This allows us to directly sample states from a pre-collected set and avoids expensive online sampling during training, significantly reducing the overall training time. 2) The state sampling process is performed before RL training, rather than during the rollout phase. This enables us to perform extensive sampling from the expert distribution, making training more scalable.

**Training approach.** Based on the step-wise RL formulation, our goal is to maximize the expected reward $R(s_t, a_t)$ shown in equation 4 according to the state distribution $d^{\pi_e}$ and $\pi_\theta$. This aligns with the approach used in RLVR methods (Guo et al., 2025b), where the policy represents a token generation process and $R(s_t, a_t)$ is the outcome reward of $\pi_\theta(a_t | s_t)$. Hence, any RL training approach can be applied to this objective $\mathcal{J}_{\text{step}}(\theta)$. For our implementation, we choose PPO (Schulman et al., 2017) as the training algorithm because of its widespread use and proven effectiveness. Specifically, suppose we expand the token generating process of $\pi_\theta(a_t | s_t)$, our PPO loss function can be defined as follows:

$$\mathcal{J}_{\text{step}}^{\text{PPO}}(\theta) = \mathbb{E}_{s \sim d^e, o_{\leq i} \sim \pi_{\theta_{\text{old}}}(\cdot | s)} \left[ \min\left( \frac{\pi_\theta(o_i \mid s, o_{<i})}{\pi_{\theta_{\text{old}}}(o_i \mid s, o_{<i})} \hat{A}_i, \; \text{clip}\left( \frac{\pi_\theta(o_i \mid s, o_{<i})}{\pi_{\theta_{\text{old}}}(o_i \mid s, o_{<i})}, 1 - \varepsilon, 1 + \varepsilon \right) \hat{A}_i \right) \right], \tag{5}$$

where $o_i$ is the $i$th token of $a_t$ and $\hat{A}_i$ is an estimator of the advantage at the token generation step $i$.

### 4.3 Agentic ML-specific reward

Having enabled efficient RL for agentic ML via the step-wise RL paradigm, the next crucial step is to convert the varied feedback into a unified, meaningful reward. While numerical metrics like validation accuracy or loss naturally serve as RL rewards, non-numerical feedback, such as compilation errors or out-of-memory failures, must be carefully incorporated to ensure the reward is coherent.

To address this, we propose an agentic ML-specific reward module that dynamically processes these diverse signals while quantifying performance improvements through scaled task-specific metrics. The key idea is to translate every execution outcome into a unified scalar value. Define $\mathcal{A}_{\text{valid}}$ as valid actions, $\mathcal{A}_{\text{edit}} \subset \mathcal{A}_{\text{valid}}$ as editing actions for ML code, $\mathcal{F}_{\text{error}}$ as error feedback (e.g., compilation failures), $\mathcal{F}_{\text{corner}}$ as corner cases (e.g., resource exhaustion), and $\mathcal{F}_{\text{success}}$ as successful executions. Let $m_t$ be the task-specific metric[1] at state $s_t$ (e.g., loss or accuracy), with $m_{\text{init}}$ and $m_{\text{best}}$ as the baseline and best human-achievable scores. The reward $R(s_t, a_t)$ is:

---

[1] We follow the official Kaggle evaluation protocol which defines a scalar metric for each ML tasks.

$$R(s_t, a_t) = \begin{cases} -1 & \text{, if } a_t \notin \mathcal{A}_{\text{valid}} \text{ or } f_t \in \mathcal{F}_{\text{error}} \\ 0 & \text{, if } a_t \in \mathcal{A}_{\text{valid}}/\mathcal{A}_{\text{edit}} \text{ or } f_t \in \mathcal{F}_{\text{corner}} \\ \frac{m_{t+1}-m_t}{m_{\text{best}}-m_{\text{init}}} & \text{, if } a_t \in \mathcal{A}_{\text{edit}} \text{ and } f_t \in \mathcal{F}_{\text{success}}. \end{cases} \quad (6)$$

This reward module handles all possible agentic ML scenarios: (1) Invalid actions or errors receive -1 to penalize faulty outputs; (2) Valid non-editing actions or corner cases receive 0 as a neutral acknowledgment of legitimacy while recognizing external constraints; (3) Success edits yield a scaled metric improvement for task-driven refinement. By unifying penalties for errors, neutrality for non-editing actions, and task-driven rewards for edits, the module provides consistent, informative feedback for iterative refinement and continuous improvement across diverse ML tasks.

## 5 EXPERIMENTS

### 5.1 EXPERIMENTAL SETUPS

**Training.** For training data collection, we adopt a GPT-4o-mini-driven (OpenAI, 2024) agent scaffolded by MLAB (Huang et al., 2023). This agent interacts with the MLAgentBench (Huang et al., 2023) agentic ML environment to generate expert trajectories. We collect 10k expert trajectories across 9 ML tasks, comprising 4 tasks from MLAgentBench and 5 from MLE-bench (Chan et al., 2024), with each trajectory limited to 15 steps and 30 minutes of runtime. Additional data collection details are provided in Appendix B. For exploration-enhanced fine-tuning, we train Qwen2.5-7B (Yang et al., 2024) using these 10k expert trajectories via supervised fine-tuning (SFT). For step-wise RL, we select 10k states sampled from expert trajectories to further train the SFT model using Proximal Policy Optimization (PPO). All training is conducted on 8 A100 GPUs. The fine-tuning stage runs for 2 epochs with a learning rate of $2e-5$, while the RL stage runs for 1 epoch with an actor learning rate of $1e-6$ and a critic learning rate of $1e-5$. See additional training details in Appendix C.1.

**Testing.** To verify the generalization ability across ML tasks of ML-Agent, we select 10 held-out tasks from MLE-bench, which are not seen during training and generally more challenging than the training tasks. Details of these tasks are provided in Appendix B.2. During testing, the MLAgentBench environment settings remain consistent with those used in training. To comprehensively assess the LLM agent's ability in autonomous ML, we propose ***Performance gain*** $\Delta_r$, the relative improvement over the initial script, defined as $\Delta_r = \beta \frac{m_{avg@8}-m_{\text{init}}}{m_{\text{init}}}$ where $m_{avg@8}$ is the mean score over 8 trajectories, $m_{\text{init}}$ is the initial script's score, and $\beta \in \{-1, 1\}$ adjusts for metrics (e.g. MAE, RMSE) to ensure positive $\Delta_r$ indicates improvement.

**Baselines.** To provide a comprehensive comparison, we evaluate ML-Agent against 3 prompted-based agentic ML methods: MLAB (Huang et al., 2023), AIDE (Jiang et al., 2025), and ML-Master (Liu et al., 2025a). All agents are tested using a diverse set of backbone LLMs, spanning small-scale open-source models (e.g., Qwen2.5-7B-Instruct (Yang et al., 2024)), medium-scale models (e.g., Qwen3-235B (Yang et al., 2025)), large-scale open-source models (e.g., DeepSeek-R1 (Guo et al., 2025b)), and state-of-the-art proprietary LLMs (Gemini-2.5-Pro (Comanici et al., 2025) and GPT-5 (OpenAI, 2025)). We keep the same time limit and number of ML code modifications for a fair comparison between agents with different scaffolds.

### 5.2 MAIN RESULTS

We conduct extensive experiments to evaluate the performance of ML-Agent, a learning-based LLM agent trained through our proposed framework for autonomous ML. Our results demonstrate that ML-Agent achieves strong and consistent performance across both held-in and held-out tasks, and exhibits continuous performance improvements during RL training.

**ML-Agent achieves superior performance across both held-in and held-out tasks.** We compare ML-Agent with 5 powerful LLM-based agents in 3 scaffolds across 3 held-in and 10 held-out tasks. As shown in Table 1, ML-Agent significantly outperforms other large open-source models, such as the 671B DeepSeek-R1. For closed-source GPT-5, our agent remains remarkably competitive. Notably, despite being trained on only 9 tasks, ML-Agent delivers top-tier results across all 10 held-out tasks, demonstrating strong generalization and effective learning from limited experience.

Table 1: Comparing 7B ML-Agent with baselines across different agent frameworks driven by proprietary/open-source LLMs on 3 **held-in** tasks (included in training) and 10 **held-out** tasks (unseen during training) from MLE-bench. For each task, we report average performance gain (%) over 8 trajectories.

| Method | Model | #Params | cifar10 | house | feedback | denoising | leaf | statoil | whale |
|---|---|---|---|---|---|---|---|---|---|
| | | | *Prompt-Based Method* | | | | | | |
| MLAB | Qwen2.5-7B-Instruct | 7B | 1.37 | 0.23 | 1.39 | 2.10 | 2.52 | -6.32 | 12.25 |
| | Qwen3-235B | 235B | 57.61 | 3.01 | 6.70 | 62.60 | -2.12 | -16.36 | 26.68 |
| | DeepSeek-R1 | 671B | 28.96 | 3.45 | 5.53 | 8.83 | 4.85 | 0.04 | 33.44 |
| | GPT-5 | N/A | 61.46 | 12.15 | 12.74 | 66.00 | -45.63 | -6.43 | 89.59 |
| | Gemini-2.5-Pro | N/A | 16.78 | 1.16 | 0.10 | 37.85 | -4.38 | -4.26 | 22.38 |
| AIDE | Qwen2.5-7B-Instruct | 7B | 11.36 | 2.42 | 7.52 | 7.33 | -4.75 | -4.33 | 0.52 |
| | Qwen3-235B | 235B | -0.10 | 2.04 | 11.10 | 41.65 | 4.75 | -2.89 | 8.26 |
| | DeepSeek-R1 | 671B | 72.55 | 5.35 | 13.07 | 33.23 | -10.25 | -4.54 | 30.77 |
| | GPT-5 | N/A | 76.53 | 22.15 | 8.77 | 77.38 | 31.50 | -9.18 | 26.42 |
| | Gemini-2.5-Pro | N/A | 53.59 | 11.13 | 9.44 | 62.72 | -84.25 | -6.08 | 56.45 |
| ML-Master | Qwen2.5-7B-Instruct | 7B | 1.03 | 0.00 | 0.10 | 2.44 | -1.38 | -3.99 | 1.12 |
| | DeepSeek-R1 | 671B | 73.43 | 18.25 | 12.07 | 14.56 | -14.75 | -2.78 | 33.39 |
| | GPT-5 | N/A | 71.64 | 22.3 | 10.54 | 10.96 | 23.88 | -2.48 | 67.07 |
| | | | *Learning-Based Method* | | | | | | |
| **ML-Agent(Ours)** | | 7B | 33.80 | 6.77 | 13.47 | 52.38 | 13.87 | 1.41 | 72.89 |

| Method | Model | #Params | learning | detecting | spooky | jigsaw | us | tabular | Avg. |
|---|---|---|---|---|---|---|---|---|---|
| | | | *Prompt-Based Method* | | | | | | |
| MLAB | Qwen2.5-7B-Instruct | 7B | 1.23 | 0.51 | -0.46 | -0.06 | 3.75 | 0.04 | 1.43 |
| | Qwen3-235B | 235B | 0.30 | 1.02 | 0.80 | 0.01 | 1.96 | -0.07 | 10.93 |
| | DeepSeek-R1 | 671B | 0.05 | 0.25 | 0.89 | 0.00 | 2.67 | -0.13 | 6.83 |
| | GPT-5 | N/A | 4.36 | 11.20 | 6.79 | 0.00 | 23.38 | 0.23 | **18.14** |
| | Gemini-2.5-Pro | N/A | 0.00 | 0.13 | 0.04 | 0.00 | 0.13 | 0.00 | 5.38 |
| AIDE | Qwen2.5-7B-Instruct | 7B | -9.78 | -0.38 | 0.07 | 0.01 | 0.00 | 0.08 | 0.77 |
| | Qwen3-235B | 235B | 2.37 | 0.43 | 0.96 | -12.15 | 0.51 | 0.00 | 4.38 |
| | DeepSeek-R1 | 671B | 1.38 | 0.31 | 0.36 | 0.01 | 5.78 | 0.14 | 11.40 |
| | GPT-5 | N/A | 4.51 | 0.13 | 4.25 | 0.14 | 29.69 | 0.11 | **20.95** |
| | Gemini-2.5-pro | N/A | 7.35 | 0.74 | 4.34 | 0.04 | 31.92 | 0.13 | 11.35 |
| ML-Master | Qwen2.5-7B-Instruct | 7B | 1.79 | 0.26 | -0.04 | -0.02 | -0.02 | 0.00 | 0.10 |
| | DeepSeek-R1 | 671B | 3.03 | 0.00 | 4.01 | -0.04 | 29.27 | 0.22 | 13.13 |
| | GPT-5 | N/A | 6.38 | 0.79 | 10.41 | 0.35 | 26.49 | 0.25 | **19.12** |
| | | | *Learning-Based Method* | | | | | | |
| **ML-Agent(Ours)** | | 7B | 1.91 | 1.74 | 1.76 | 0.01 | 12.96 | 0.20 | **16.40** |

**ML-Agent efficiently achieves good performance with much lower cost.** As illustrated in Figure 2, we plot the average performance gain against the average cost per trajectory for various agents. Our proposed ML-Agent (the star) is a clear outlier, positioned in the optimal top-left corner. It achieves highly competitive performance gain of over 15% while maintaining an exceptionally low cost of less than 0.01$ per trajectory. In contrast, baseline agents like MLAB using powerful models such as GPT-5 incur costs that are more than 20 times higher for similar or even lower performance. This result highlights the significant efficiency of learning-based paradigm, proving it can produce a state-of-the-art agent without relying on expensive, large-scale models.

**ML-Agent achieves continuous performance improvements.** Figure 3 shows that ML-Agent demonstrates consistent performance improvement across both held-in and held-out tasks as training progresses. This highlights the effectiveness of our step-wise RL paradigm and exploration-enriched fine-tuning in enabling continuous learning from ML environmental feedback, ultimately allowing ML-Agent to outperform all baseline methods.

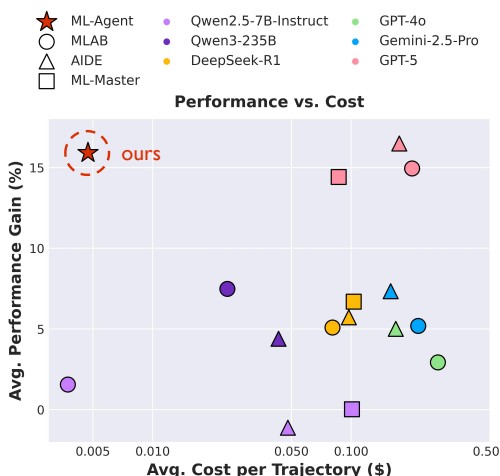

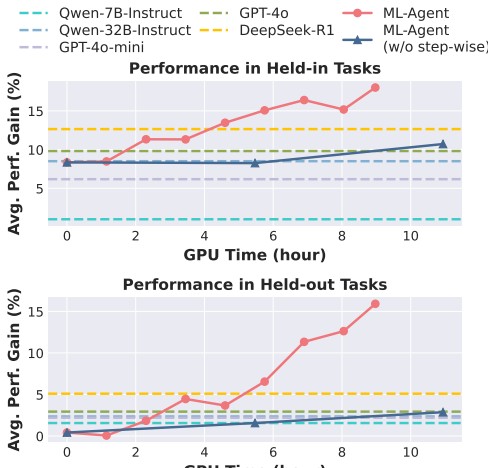

Figure 2: Comparison of average performance gain (%) vs. cost ($) across different models and scaffolds on 10 held-out tasks. Our ML-Agent significantly outperforms other baselines with a competitive gain at a lower cost.

Figure 3: ML-Agent achieves continuous performance improvements; Step-wise RL (evaluated every 5 steps) is more efficient than episode-wise RL (evaluated every 1 step) on both held-in and held-out tasks.

## 5.3 ANALYSIS

**Exploration-enriched fine-tuning is crucial for step-wise RL training.** To validate the efficacy of exploration-enriched fine-tuning in enhancing subsequent RL training, we replace our exploration-enriched fine-tuned model (ML-Agent-SFT) with Qwen2.5-7B (Qwen-7B-Base), Qwen2.5-7B-Instruct (Qwen-7B-Instruct), and DeepSeek-R1-Distill-Qwen-7B (Guo et al., 2025b)(Qwen-7B-Distill) as base models for the RL training. We evaluate the average performance gain of the resulting RL-trained agents on held-in and held-out tasks (Figure4). The agent trained from Qwen-7B-Distill fails to generate valid actions due to distillation-induced format issues, resulting in ineffective

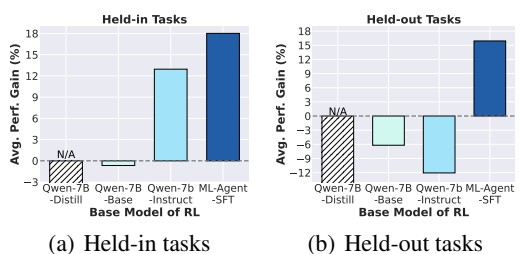

Figure 4: Exploration-enriched fine-tuning is crucial for RL training. "N/A" means the training based on the model fails to generate valid results.

learning. The agent trained from Qwen-7B-Base shows overall performance degradation from limited instruction-following capabilities. The agent trained from Qwen-7B-Instruct achieves +13% gains on held-in tasks but -12% on held-out tasks, indicating poor generalization. In contrast, the agent trained from our ML-Agent-SFT achieves +18% and +16% improvement on held-in and held-out tasks, respectively, with greater action diversity during autonomous ML experimentation (Figure 7). These results confirm that exploration-enriched fine-tuning promotes format-compliant, diverse actions, enhancing exploration and generalization in step-wise RL.

**Effectiveness of Step-wise RL Training.** To improve training efficiency and scalability, we propose a step-wise RL approach that samples single states from expert trajectories and evaluates atomic actions. To validate this, we implement an alternative episode-wise RL approach, where the policy rolls out the entire trajectory from the task description during data collecting phase in RL. Both methods are initialized from the same ML-Agent-SFT model and trained for 39 steps. We measure GPU time every 5 steps for step-wise RL and 1 step for episode-wise RL. As shown in Figure 3, step-wise RL adapts more quickly and achieves faster performance gains on both held-in and held-out tasks, while the performance of episode-wise RL improves slowly and incurs much higher time cost. These results demonstrate that step-wise RL not only improves training efficiency by avoiding expensive online rollouts, but also leads to improved performance through targeted single-step updates.

Table 2: Ablation study on the ML-specific reward module, indicating the necessity of three components. The three components are normalized performance reward ($R_{\text{perf.}}$), format reward ($R_{\text{format}}$), and corner cases reward ($R_{\text{corner}}$). We report the average performance gain (%) for each task.

| | $R_{\text{perf.}}$ | Task $R_{\text{format}}$ | $R_{\text{corner}}$ | cifar10 | house | feedback | leaf | detecting | us | tabular | whale |
|---|---|---|---|---|---|---|---|---|---|---|---|
| ① | ✗ | ✓ | ✓ | 17.58 | 3.94 | 7.79 | 4.75 | 0.26 | 6.40 | -24.96 | 23.24 |
| ② | ✓ | ✗ | ✓ | 10.98 | 6.17 | 8.34 | -30.25 | 0.03 | 6.27 | -12.54 | 2.84 |
| ③ | ✓ | ✓ | ✗ | 13.56 | 6.64 | 7.67 | 8.50 | 0.58 | 8.67 | -0.48 | 28.06 |
| ④ | ✓ | ✓ | ✓ | 33.80 | 6.77 | 13.47 | 13.87 | 1.74 | 12.96 | 0.20 | 72.89 |

**Effectiveness of agentic-ML specific reward module.** Ablation studies in Table 2 show each reward component is essential: (1) Performance($R_{\text{perf.}}$): Replacing the scaled performance difference with binary reward leads to noticeable performance drops. This confirms that fine-grained reward signals are more informative for learning meaningful improvements. (2) Format($R_{\text{format}}$): Removing format constraints causes the largest degradation (e.g., -11.75% on cifar-10), emphasizing the necessity of syntactic and semantic correctness of agent's output format. (3) Corner cases($R_{\text{corner}}$): Disabling the neutral reward for corner cases has minimal impact due to their rarity, but improves training stability by preventing over-penalization of non-fatal issues. In summary, each component of the reward module plays a distinct and complementary role: $R_{\text{perf.}}$ drives performance improvement, $R_{\text{format.}}$ ensures actions validity, and $R_{\text{corner}}$ maintains robustness under real-world limitations. Together, they form a coherent and comprehensive reward structure during RL training for agentic ML.

**Effects of task numbers in RL.** We investigate the impact of using different numbers of ML tasks (0, 3, 6, 9) during step-wise RL training, where the "0 task" condition corresponds to ML-Agent-SFT. We evaluate performance in terms of average performance gain on held-in and held-out tasks. As shown in Figure 5, performance on both task types improves monotonically as the number of ML tasks increases during RL training. Specifically, training with 3, 6, and 9 ML tasks using step-wise RL lifts the average performance gain on held-out tasks from nearly 0% to approximately 3%, 6%, and 16%, respectively. These results indicate that expanding the diversity of ML tasks during RL not only refines the agent's ability on familar tasks but also significantly improves the agent's ability to generalize across unseen tasks.

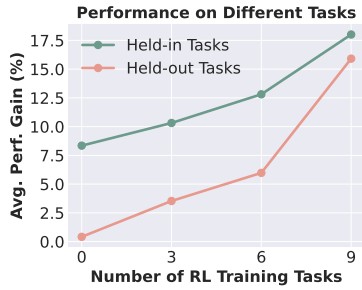

Figure 5: Effects of training task number on RL performance. While the pure sft model shows minimal generalization, RL drives generalization.

**Case study.** To provide an intuitive understanding, we present several examples in the Appendix C.3, demonstrating task specifications, initial code implementations, baseline and our model's execution trajectories. These demonstrate that our methodology: (1) generates diverse action sequences through comprehensive reasoning, (2) automatically initiates backtracking mechanisms when performance metrics remain unimproved by proposed modifications, and (3) maintains operational effectiveness on novel tasks through generalized reasoning capabilities.

## 6 CONCLUSION

In this paper, we propose a training framework for an LLM-based agent on autonomous machine learning tasks. Unlike heuristic prompt-based methods, our method enables agents to learn from task-solving experiences, iteratively refine strategies, and generalize across tasks. The framework involved exploration-enriched fine-tuning, efficient step-wise RL training, and agentic ML-specific reward module. Extensive experiments demonstrate that ML-Agent, powered by a 7B-parameter LLM, surpasses agents using 671B models and achieves state-of-the-art performance on 13 tasks, including cross-task generalization. This work advances autonomous ML engineering from rule-based automation to dynamic, experience-driven learning, reducing reliance on human intervention.

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

## A   PROBLEM FORMULATION

**Reformulation from equation 1 to equation 3.** Starting from equation 1, suppose the initial state distribution is $d^{\pi_\theta}(s_0)$, the state transition probability is $p_\pi(s_{t+1}|s_t, a_t)$, then we have

$$\mathcal{P}_{\pi_\theta}(\tau) = d^{\pi_\theta}(s_0) \prod_{t=0}^{n-1} p(s_{t+1}|s_t, a_t)\pi_\theta(a_t|s_t). \tag{7}$$

Hence the reformulation is:

$$\begin{aligned}
\mathcal{J}(\theta) &= \mathbb{E}_{\pi_\theta}[R(\tau)] \\
&= \sum_{\tau} \mathcal{P}_{\pi_\theta}(\tau)R(\tau) \\
&= \sum_{s_0,a_0,\ldots,s_n} \left( d^{\pi_\theta}(s_0) \prod_{t=0}^{n-1} p(s_{t+1}|s_t, a_t)\pi_\theta(a_t|s_t) \right) \left( \sum_{t=0}^{n} R(s_t, a_t) \right) \\
&= \sum_{t=0}^{n-1} \sum_{s_0,a_0,\ldots,s_n} \left( d^{\pi_\theta}(s_0) \prod_{k=0}^{n-1} p(s_{k+1}|s_k, a_k)\pi_\theta(a_k|s_k) \right) R(s_t, a_t) \\
&= \sum_{t=0}^{n-1} \sum_{s_t,a_t} \left( \sum_{s_0,a_0,\ldots s_{t-1},a_{t-1}} d^{\pi_\theta}(s_0) \prod_{k=0}^{t-1} p(s_{k+1}|s_k, a_k)\pi_\theta(a_k|s_k) \right) \pi_\theta(a_t|s_t)R(s_t, a_t)
\end{aligned} \tag{8}$$

However, we can define the state distribution $d^{\pi_\theta}(s_t)$ as the probability agent visits state $s_t$ at time $t$. Then according to this definition, this probability can be written as:

$$d^{\pi_\theta}(s_t) = \sum_{s_0,a_0,\ldots s_{t-1},a_{t-1}} d^{\pi_\theta}(s_0) \prod_{k=0}^{t-1} p(s_{k+1}|s_k, a_k)\pi_\theta(a_k|s_k). \tag{9}$$

Then we have

$$\begin{aligned}
\mathcal{J}(\theta) &= \sum_{t=0}^{n-1} \sum_{s_t,a_t} d^{\pi_\theta}(s_t)\pi_\theta(a_t|s_t)R(a_t, s_t) \\
&= \sum_{t=0}^{n-1} \sum_{s_t \in \mathcal{S}} d^{\pi_\theta}(s_t) \sum_{a_t \in \mathcal{A}} \pi_\theta(a_t|s_t)R(a_t, s_t) \\
&= \mathcal{J}_{\text{step}}(\theta)
\end{aligned} \tag{10}$$

## B   MACHINE LEARNING TASKS AND DATA COLLECTION PROCESS

### B.1   DATA COLLECTING PIPELINE FOR EXPLORATION-ENRICHED FINE-TUNING

We construct diverse action pools along three semantic axes—**Data**, **Model**, and **Learning**—to support structured exploration. For each axis, we prompt a frozen LLM (GPT-4o-mini) to generate a large set of candidate actions (e.g., "Add MixUp augmentation", "Switch to AdamW optimizer"). To promote diversity, we embed all candidates using a sentence transformer and apply farthest-point sampling (FPS) to select a compact, representative subset. The resulting pools $\mathcal{P}_{\text{Data}}$, $\mathcal{P}_{\text{Model}}$, and $\mathcal{P}_{\text{Learning}}$ are fixed during training.

During data collection, we form exploration-enriched prompts by randomly selecting 1–3 axes, shuffling their order, and drawing one action from each corresponding pool. These actions are concatenated into an initial instruction for the expert agent, which then interacts with a fast-executable ML environment (e.g., small-scale tabular or vision tasks) to produce a full trajectory. The complete pipeline is summarized in Algorithm 1.

### B.2   DETAILS OF MACHINE LEARNING TASKS

The machine learning tasks utilized in our paper are all from MLAgentBench or MLE-bench. Table 3 shows all 9 training tasks and 10 testing tasks. The 9 training tasks contain 4 tasks from

---

**Algorithm 1** Exploration-Enriched Trajectory Generation

---

**Require:** Semantic axes $\mathcal{X} = \{\text{Data}, \text{Model}, \text{Learning}\}$,
      Set of fast-executable tasks $\mathcal{N}$, each with base description $p_n^{\text{task}}$
**Ensure:** Dataset of expert trajectories $\mathcal{D}$
 1: #Phase 1: Build diverse action pools via FPS
 2: **for** each axis $X \in \mathcal{X}$ **do**
 3:    Generate $M$ candidate actions $\mathcal{C}_X$ using LLM prompting
 4:    $\mathcal{P}_X \leftarrow \text{FARTHESTPOINTSAMPLING}(\mathcal{C}_X, K)$ {Select $K$ diverse actions}
 5: **end for**
 6: #Phase 2: Generate trajectories
 7: $\mathcal{D} \leftarrow \emptyset$
 8: **for** each task $n \in \mathcal{N}$ **do**
 9:    Sample $k \sim \text{Uniform}\{1, 2, 3\}$
10:    Sample $k$ distinct axes $\{X_1, \ldots, X_k\} \subset \mathcal{X}$
11:    Sample $a_i \sim \text{Uniform}(\mathcal{P}_{X_i})$ for $i = 1, \ldots, k$
12:    Form prompt: $p_n \leftarrow p_n^{\text{task}}.\text{format}(a_1, \ldots, a_k)$
13:    Run expert LLM (GPT-4o-mini) on task $n$ with prompt $p_n$
14:    Record trajectory $\tau$
15:    $\mathcal{D} \leftarrow \mathcal{D} \cup \{\tau\}$
16: **end for**
17: **return** $\mathcal{D}$

---

MLAgentBench and 5 from MLE-bench (Chan et al., 2024); while the 10 testing tasks are all from MLE-bench.

The selection strategy of training tasks aims to enhance data collection efficiency. Specifically, we select relatively simpler machine learning tasks (e.g. tasks labeled with low complexity in MLE-bench) for training. These training tasks typically involve smaller datasets, which enable faster iterations. For testing, we select relatively more complex tasks to evaluate the generalization capability. In addition, the training tasks and test tasks span three machine learning data types (image, text and tabular) and two general task categories (regression and classification).

Specifically, Each task consists of the following components: (1) training, validation, and test data; (2) an initial bug-free script, `train.py`, generated by GPT-4o-mini; (3) an evaluation script, `eval.py`, which is used to calculate the test score from the submitted results; (4) a problem description file, `research_problem.txt`; and (5) a `prepare.py` script to download the data if necessary. An example file structure and related problem descriptions are shown in Figure 6. To ensure clarity regarding the task details and training objectives, we have refined some initial prompts from MLAgentBench by incorporating specific targets, such as "try your best to increase the test accuracy to 99.99%" (see in the right box in Figure 6). The format for the initial prompt, including the tool and format prompts, follows actions defined by MLAgentBench (see Table 5).

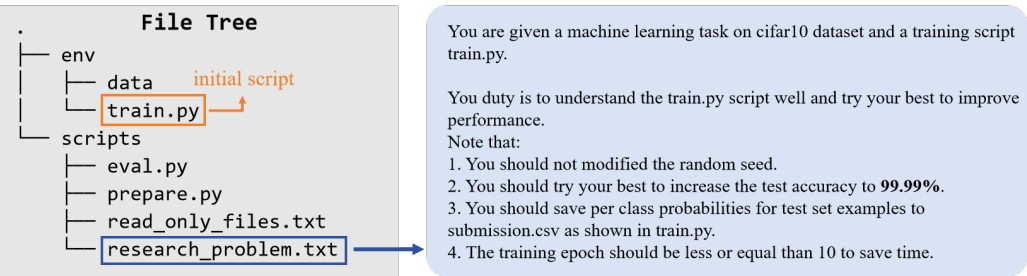

Figure 6: Task file structure and prompt about the machine learning problem of cifar-10 task, for instance.

Table 3: All training and testing tasks used in our experiments. MLA and MLE stand for MLAgent-Bench (Huang et al., 2023) and MLE-bench (Chan et al., 2024) respectively.

| Task Name | Data Type | Task Type | Metric | Source |
|---|---|---|---|---|
| **Training** | | | | |
| cifar-10 | Image | Classification | Acc. (%) ↑ | MLA |
| aerial-cactus-identification | Image | Classification | AUC ↑ | MLE |
| dogs-vs-cats-redux-kernels-edition | Image | Classification | Logloss ↓ | MLE |
| plant-pathology-2020-fgvc7 | Image | Classification | AUC ↑ | MLE |
| home-data-for-ml-course | Tabular | Regression | MAE ↓ | MLA |
| spaceship-titanic | Tabular | Regression | Acc. (%) ↑ | MLA |
| nomad2018-predict-transparent-conductors | Tabular | Regression | RMSLE ↓ | MLE |
| feedback-prize-english-language-learning | Text | Classification | MCRMSE ↓ | MLA |
| ogbn-arxiv (Maas et al., 2011) | Graph | Classification | Acc. (%) ↑ | MLA |
| **Testing** | | | | |
| denoising-dirty-documents | Image | Generation | RMSE ↓ | MLE |
| leaf-classification | Image | Classification | Logloss ↓ | MLE |
| statoil-iceberg-classifier-challenge | Image | Classification | Logloss ↓ | MLE |
| whale-categorization-playground | Image | Classification | MAP@5 ↑ | MLE |
| learning-agency-lab-automated-essay-scoring-2 | Text | Regression | QWK ↑ | MLE |
| detecting-insults-in-social-commentary | Text | Classification | Acc. (%) ↑ | MLE |
| spooky-author-identification | Text | Classification | Logloss ↓ | MLE |
| jigsaw-toxic-comment-classification-challenge | Text | Classification | AUC ↑ | MLE |
| us-patent-phrase-to-phrase-matching | Tabular | Regression | PCC ↑ | MLE |
| tabular-playground-series-dec-2021 | Tabular | Regression | Acc. (%) ↑ | MLE |

Table 4: Actions in MLAgentBench (Huang et al., 2023), where each action has a name, input and output. Most of the actions are primitive actions that include file system operations and python script execution. The last two are compound actions that is composed of multiple primitive actions and LM calls.

| Action Name | Input | Observation |
|---|---|---|
| List Files | directory (e.g. `.`) | list of files in the directory |
| Copy File | Source (e.g. `train.py`), destination (e.g. `train_copy.py`) | A success or error message |
| Inspect Script Lines | file name, start line number, end line number | the file content between start and end line numbers |
| Execute Script | file name (e.g. `train.py`) | Any output from the execution |
| Final Answer | None | None |
| Understand File | file name, a query (e.g. the model architecture) | retrieved content from the file relevant to the query |
| Edit Script | file name, edit instruction (e.g. change epoch to 20), save file name | The diff of the edited file based on the instruction |

Table 5: Initial prompt template for agents on autonomous machine learning.

You are a helpful research assistant. You have access to the following tools:
**{tools_prompt}**
Research Problem: **{research_problem}**
Always respond in this format exactly:
**{format_prompt}**
Observation:
,,,

the result of the action
,,,

**Tools prompt (`{tools_prompt}`) in initial prompt.**

```
You are a helpful research assistant. You have access to the following tools:
- List Files:
        Use this to navigate the file system.
        Usage:
        ```
        Action: List Files
        Action Input: {
            "dir_path": [a valid relative path to a directory, such as "." or "folder1/
                folder2"]
        }
        Observation: [The observation will be a list of files and folders in dir_path or
             current directory is dir_path is empty, or an error message if dir_path is
             invalid.]
        ```

- Copy File:
        Use this to copy a file to a new location with a new name.
        Usage:
        ```
        Action: Copy File
        Action Input: {
            "source": [a valid file name with relative path to current directory if needed],
            "destination": [a valid file name with relative path to current directory if
                needed]
        }
        Observation: [A success message if the file is copied successfully, or an error
             message if the file cannot be copied.]
        ```

- Execute Script:
        Use this to execute the python script. The script must already exist.
        Usage:
        ```
        Action: Execute Script
        Action Input: {
            "script_name": [a valid python script name with relative path to current
                directory if needed]
        }
        Observation: [The observation will be output of the script or errors.]
        ```

- Final Answer:
        Use this to provide the final answer to the current task.
        Usage:
        ```
        Action: Final Answer
        Action Input: {
            "final_answer": [a detailed description on the final answer]
        }
        Observation: [The observation will be empty.]
        ```

- Understand File:
        Use this to read the whole file and understand certain aspects. You should provide
             detailed description on what to look for and what should be returned. To get a
             better understanding of the file, you can use Inspect Script Lines action to
             inspect specific part of the file.
        Usage:
        ```
        Action: Understand File
        Action Input: {
            "file_name": [a valid file name with relative path to current directory if
                needed],
            "things_to_look_for": [a detailed description on what to look for and what
                should returned]
        }
        Observation: [The observation will be a description of relevant content and lines in
             the file. If the file does not exist, the observation will be an error message
             .]
        ```

- Inspect Script Lines:
        Use this to inspect specific part of a python script precisely, or the full content
             of a short script. The number of lines to display is limited to 100 lines. This
             is especially helpful when debugging.
        Usage:
        ```
        Action: Inspect Script Lines
        Action Input: {
```

```
                    "script_name": [a valid python script name with relative path to current
                         directory if needed],
                    "start_line_number": [a valid line number],
                    "end_line_number": [a valid line number]
             }
             Observation: [The observation will be the content of the script between
                    start_line_number and end_line_number . If the script does not exist, the
                    observation will be an error message.]
             '''

- Edit Script (AI):
             Use this to do a relatively large but cohesive edit over a python script. Instead of
                    editing the script directly, you should describe the edit instruction so that
                    another AI can help you do this.
             Usage:
             '''
             Action: Edit Script (AI)
             Action Input: {
                    "script_name": [a valid python script name with relative path to current
                         directory if needed. An empty sctipt will be created if it does not exist
                         .],
                    "edit_instruction": [a detailed step by step description on how to edit it.],
                    "save_name": [a valid file name with relative path to current directory if
                         needed]
             }
             Observation: [The observation will be the edited content of the script. If the
                    script does not exist, the observation will be an error message. You should
                    always double check whether the edit is correct.]
             '''
```

Table 6: Response format requirement (`{format_prompt}`) in the initial prompt.

Reflection: What does the observation mean? If there is an error, what caused the error and how to debug?
Research Plan and Status: The full high-level research plan, with current status and confirmed results of each step briefly annotated. It must only include progress that has been made by previous steps. If there is any update, enclose the new update text in double asterisks `**like this**`. If there is no update, just copy the previous step Research Plan and Status. The high-level plan from the previous step should be fully retained, unless it is intentionally revised.
Fact Check: List all objective statements in the updates to Research Plan and Status one by one and point out whether it is guessed versus directly confirmed by the previous observation directly above. Performance numbers can only be confirmed by running the code and observing the output.
Thought: What you are currently doing, what actions to perform and why
Action: The action to take, should be one of the names of the tools
Action Input: The input to the action as a valid JSON string

## B.3 DETAILS OF DATA COLLECTION

In this paper, we use the MLAgentBench (Huang et al., 2023) environment to collect training trajectories across 9 machine learning tasks. The environment needs an LLM-based agent to take actions and send feedback to the agent. This will iterate for certain steps. We employ GPT-4o-mini (OpenAI, 2024) as the LLM-based agent to generate thinking and action following Table B.2. This agent interacts with the environment, while Qwen2.5-Coder-32B-Instruct (Yang et al., 2024) powers the coder agent, which is responsible for writing code and understanding files within the environment.

Each trajectory comprises a multi-turn conversation between the agent and the environment. For each trajectory, we set the maximum number of steps as 15 and the time limit as 30 minutes to control the length and duration of interactions. Finally, we generated 10k trajectories on 9 tasks. These trajectories are utilized both in SFT training and PPO training.

Since each task in the MLAgentBench environment requires an initial script, tasks sourced from MLE-bench do not have a natural initial script. To address this, we generate simple, bug-free initial scripts for those tasks using GPT-4o-mini to meet the environment's requirements.

To diversify the trajectories we collect for SFT training, we curate an initial idea pool of at least 100 diverse ideas which may potentially improve the performance of our initial script. We calculate the embedding distance of each idea in initial idea pool and filter out the top 10 initial ideas whose average embedding distance is farthest to others. These ideas form a defined idea pool, which guides the first step of each trajectory. For the generation of each trajectory, we randomly select 1 to 3 idea combinations from this idea pool and prioritize their implementation in the initial step by including the relevant instructions in the file `research_problem.txt` (see Figure 6). Table 7 show the prompt we use and Table 8 shows an example of defined idea pool for the first step.

Table 7: The prompt we use to generate the data-preprocessing idea pool.

> You are given a machine learning task and an initial script on the task.
>
> The machine learning task description is:
> {task_description}
>
> The initial script is:
> {initial_script}
>
> You should give {number_to_generate} advice that may potentially improve the metric performance(e.g. accuracy) of the script on this machine learning task. Your advice can only be related to data preprocessing.
> The advice in your answer should strictly follow the following format(one advice should be in a line), note that [advice] flag should be mentioned only once in your answer:
> [advice]
> YOUR ADVICE HERE
> ...

Table 8: An example of the first step action space(after filtering) when collecting training trajectories.

> Tune the momentum parameter in the optimizer for better convergence.
> Use early stopping to terminate training when the test accuracy starts decreasing.
> Experiment with focal loss to deal with imbalanced data if classes are not evenly distributed.
> Regularize model weights with L1 or L2 regularization.
> Implement feature visualization to understand what features are being learned.
> Use a higher resolution for input images, if feasible, to capture more details.
> Increase the complexity of the neural network by adding more convolutional layers.
> Explore semi-supervised learning methods to leverage unlabeled data for training improvements.
> Normalize the data further by scaling the input images to a range of [0, 1].
> Experiment with different batch sizes to see if a smaller or larger batch size affects performance.

## C EXPERIMENTAL DETAILS

### C.1 DETAILS OF EXPERIMENTAL SET-UP

**Training details.** We implement our supervised fine-tuning (SFT) and proximal policy optimization (PPO) training using 8 A100s. For the SFT, the code base is LLama-Factory (Zheng et al., 2024),

where we fully fine-tune the qwen2.5-7b model for 2 epochs with batch size 64 and learning rate $2e - 5$. For the PPO, the code base is VeRL (Sheng et al., 2024). The PPO training setup involves the following hyperparameters and configurations: the training batch size is set to 256, and the number of epochs is 1. Additionally, the learning rate of actor and critic is set as $1e - 6$ and $1e - 5$, respectively, and the coefficient of KL is 0.001.

**Baseline details.** We show the specific versions of baselines in Table 9.

Table 9: Model Version and Identifier Mapping

| Model Name | Version |
|---|---|
| GPT-4o-mini | GPT-4o-mini-2024-07-18 |
| GPT-4o | GPT-4o-2024-08-06 |
| Qwen-7B-Base | Qwen2.5-7B |
| Qwen-7B-Instruct | Qwen2.5-7B-Instruct |
| Qwen-32B-Instruct | Qwen2.5-32B-Instruct |

## C.2 ADDITIONAL ABLATION STUDY

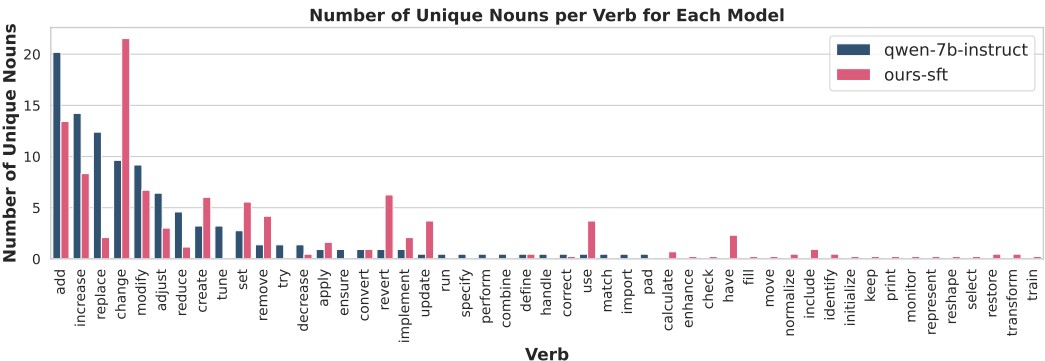

Figure 7: Unique noun counts per verb across 100 randomly sampled edit instructions, comparing the Qwen2.5-7B-Instruct model (blue) with the ML-Agent-SFT model (red).

**Diversity.** Figure 7 compares the number of unique nouns associated with each editing verb in two models: Qwen2.5-7B-Instruct and ours-sft (ML-Agent-SFT). To generate these counts, we randomly sampled 100 edit_instruction sentences from the recorded expert trajectories. Then, we utilize an open-source NLP toolkit SpaCy to obtain the verb and noun for each edit_instruction sentence. Results show that after supervised fine-tuning with expert's trajectories, the model can output a broader variety of actions, evidenced by the higher counts of unique nouns per verb.

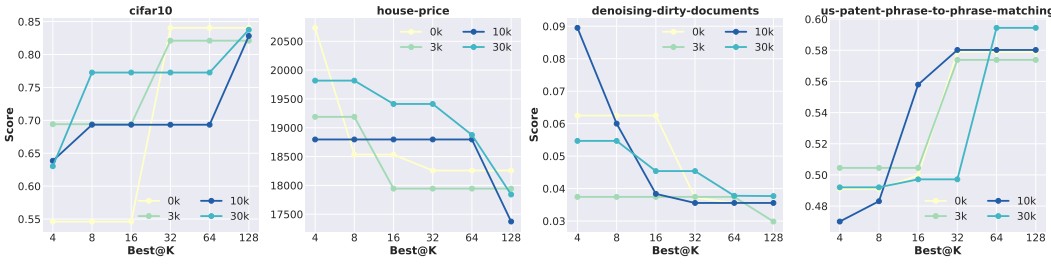

Figure 8: Different number of training samples in RL, starting from our sft model.

**Effects of training data size in RL.**

Here, we investigate how varying the number of training data samples (0k, 3k, 10k, 30k) affects the performance in RL. The 0k scenario represents ML-Agent-SFT model. For every model, we run 128 trajectories for each task and evaluate the **best@K**, where K ranges over [4, 8, 16, 32, 64, 128], as shown in Figure 8. In particular, for the two held-out tasks (second row), training with ppo (3k, 6k, and 9k) improves performance faster than 0k as the number of sampled trajectories increases.

**Is thought helpful?** In Table 10, we compare the performance of models with and without the requirement of thought before taking action across 13 tasks. The thought process includes several key components such as "Reflection," "Research Plan and Status," "Fact Check," "Thought," "Action," and "Action Input,". In contrast, the model without thought only requires "Action" and "Action Input." Note that the supervised fine-tuning data is also consistent with the key requirement. The models with thought generally exhibit higher improved performance on both held-in and held-out tasks. For instance, on the held-in cifar-10, the model with thought reaches 33.80% performance gain compared to 13.03% when thought is not required. This trend continues across the held-out tasks, where the model with thought shows higher accuracy and lower loss, demonstrating the importance of incorporating a thoughtful reflection and planning phase for Autonomous machine learning via RL.

Table 10: Performance comparison of reinforcement learning models with and without the requirement of thought prior to action. Average performance gains (%) are shown for both held-in and held-out tasks, highlighting improvements in various tasks when thought is incorporated.

| Thought? | Held-in tasks | | | Held-out tasks | | | | | | | | |
|---|---|---|---|---|---|---|---|---|---|---|---|---|
| | cifar-10 | house. | feedback | denoising. | leaf. | statoil. | learning. | detecting. | spooky. | jigsaw. | us. | tabular. |
| ✗ | 13.03 | 5.68 | 9.88 | 28.66 | 2.50 | -0.03 | 1.27 | 0.64 | -3.40 | 0.00 | 7.15 | -0.02 |
| ✓ | 33.80 | 6.77 | 13.47 | 52.38 | 13.87 | 1.41 | 1.91 | 1.74 | 1.76 | 0.01 | 12.96 | 0.20 |

## C.3 CASE STUDY

In this section, we will present more detailed case study on trajectories generated by ML-Agent(ppo) on some test tasks such as `denoising-dirty-documents`. In Appendix C.3.1, we show the task description for `denoising-dirty-documents`. In Appendix C.3.2, we show the initial script for `denoising-dirty-documents`. In Appendix C.3.3, we show partial trajectory generated by ML-Agent. We give an analysis in Appendix C.3.4.

### C.3.1 TASK DESCRIPTION FOR `denoising-dirty-documents`

Task description for `denoising-dirty-documents`

```
You are given a machine learning task on "denoising-dirty-documents" dataset. The dataset descriptions are given below:

# Description

[Optical Character Recognition](http://en.wikipedia.org/wiki/Optical_character_recognition) (OCR) is the process of getting type or
    handwritten documents into a digitized format. If you've read a classic novel on a digital reading device or had your doctor
    pull up old healthcare records via the hospital computer system, you've probably benefited from OCR.

OCR makes previously static content editable, searchable, and much easier to share. But, a lot of documents eager for digitization
    are being held back. Coffee stains, faded sun spots, dog-eared pages, and lot of wrinkles are keeping some printed documents
    offline and in the past.

This competition challenges you to give these documents a machine learning makeover. Given a dataset of images of scanned text that
    has seen better days, you're challenged to remove the noise. Improving the ease of document enhancement will help us get that
    rare mathematics book on our e-reader before the next beach vacation.

We've kicked off the fun with a few [handy scripts to get you started on the dataset](https://www.kaggle.com/c/denoising-dirty-
    documents/scripts).

# Evaluation

Submissions are evaluated on the [root mean squared error](https://www.kaggle.com/wiki/RootMeanSquaredError) between the cleaned
    pixel intensities and the actual grayscale pixel intensities.

# Submission File

Form the submission file by melting each images into a set of pixels, assigning each pixel an id of image_row_col (e.g. 1_2_1 is
    image 1, row 2, column 1). Intensity values range from 0 (black) to 1 (white). The file should contain a header and have the
    following format:
```

```
```
id,value1_1_1,1
1_2_1,1
1_3_1,1
etc.
```

## Dataset Description

You are provided two sets of images, train and test. These images contain various styles of text, to which synthetic noise has been
    added to simulate real-world, messy artifacts. The training set includes the test without the noise (train_cleaned). You must
    create an algorithm to clean the images in the test set.
```

### C.3.2  INITIAL SCRIPT FOR denoising-dirty-documents

Initial script for denoising-dirty-documents

```python
import os
import numpy as np
import pandas as pd
from PIL import Image
import glob
from sklearn.model_selection import train_test_split
import torch
from torch.utils.data import Dataset, DataLoader
from torchvision import transforms
import torch.nn as nn
import torch.optim as optim
import torch.nn.functional as F
import math

# Set device
device = torch.device("cuda" if torch.cuda.is_available() else "cpu")

# Define dataset
class DenoisingDataset(Dataset):
    def __init__(self, noisy_images, clean_images=None, transform=None):
        self.noisy_images = noisy_images
        self.clean_images = clean_images
        self.transform = transform

    def __len__(self):
        return len(self.noisy_images)

    def __getitem__(self, idx):
        noisy_image = Image.open(self.noisy_images[idx]).convert("L")
        if self.transform:
            noisy_image = self.transform(noisy_image)
        if self.clean_images is not None:
            clean_image = Image.open(self.clean_images[idx]).convert("L")
            if self.transform:
                clean_image = self.transform(clean_image)
            return noisy_image, clean_image
        else:
            return noisy_image

# Custom collate function to pad images to the same size
def collate_fn(batch):
    if len(batch[0]) == 2:
        imgs, targets = zip(*batch)
    else:
        imgs = batch
        targets = None

    # Compute necessary heights and widths after padding to next multiple of 8
    heights = []
    widths = []
    for img in imgs:
        c, h, w = img.shape
        new_h = ((h - 1) // 8 + 1) * 8
        new_w = ((w - 1) // 8 + 1) * 8
        heights.append(new_h)
        widths.append(new_w)

    max_h = max(heights)
    max_w = max(widths)

    padded_imgs = []
    if targets is not None:
        padded_targets = []

    for i, img in enumerate(imgs):
        c, h, w = img.shape
        pad_h = max_h - h
        pad_w = max_w - w
        padding = (0, pad_w, 0, pad_h) # left, right, top, bottom
        padded_img = F.pad(img, padding)
        padded_imgs.append(padded_img)
        if targets is not None:
            target = targets[i]
```

```
                padded_target = F.pad(target, padding)
                padded_targets.append(padded_target)

        imgs_tensor = torch.stack(padded_imgs, dim=0)
        if targets is not None:
            targets_tensor = torch.stack(padded_targets, dim=0)
            return imgs_tensor, targets_tensor
        else:
            return imgs_tensor

# Load image paths
noisy_images = sorted(glob.glob("./train/*.png"))
clean_images = sorted(glob.glob("./train_cleaned/*.png"))

# Split into train and validation sets
train_noisy, val_noisy, train_clean, val_clean = train_test_split(
    noisy_images, clean_images, test_size=0.2, random_state=42
)

# Define transforms
transform = transforms.Compose(
    [
        transforms.ToTensor(),
    ]
)

# Create datasets and dataloaders
train_dataset = DenoisingDataset(train_noisy, train_clean, transform)
val_dataset = DenoisingDataset(val_noisy, val_clean, transform)

train_loader = DataLoader(
    train_dataset, batch_size=4, shuffle=True, collate_fn=collate_fn
)
val_loader = DataLoader(val_dataset, batch_size=4, shuffle=False, collate_fn=collate_fn)

# Define the model (modified U-Net with cropping)
class UNet(nn.Module):
    def __init__(self):
        super(UNet, self).__init__()
        # Encoder
        self.enc1 = nn.Sequential(
            nn.Conv2d(1, 64, 3, padding=1),
            nn.ReLU(),
            nn.Conv2d(64, 64, 3, padding=1),
            nn.ReLU(),
        )
        self.pool1 = nn.MaxPool2d(2, 2)

        self.enc2 = nn.Sequential(
            nn.Conv2d(64, 128, 3, padding=1),
            nn.ReLU(),
            nn.Conv2d(128, 128, 3, padding=1),
            nn.ReLU(),
        )
        self.pool2 = nn.MaxPool2d(2, 2)

        self.enc3 = nn.Sequential(
            nn.Conv2d(128, 256, 3, padding=1),
            nn.ReLU(),
            nn.Conv2d(256, 256, 3, padding=1),
            nn.ReLU(),
        )
        self.pool3 = nn.MaxPool2d(2, 2)

        # Bottleneck
        self.bottleneck = nn.Sequential(
            nn.Conv2d(256, 512, 3, padding=1),
            nn.ReLU(),
            nn.Conv2d(512, 512, 3, padding=1),
            nn.ReLU(),
        )

        # Decoder
        self.up3 = nn.ConvTranspose2d(512, 256, 2, stride=2)
        self.dec3 = nn.Sequential(
            nn.Conv2d(512, 256, 3, padding=1),
            nn.ReLU(),
            nn.Conv2d(256, 256, 3, padding=1),
            nn.ReLU(),
        )

        self.up2 = nn.ConvTranspose2d(256, 128, 2, stride=2)
        self.dec2 = nn.Sequential(
            nn.Conv2d(256, 128, 3, padding=1),
            nn.ReLU(),
            nn.Conv2d(128, 128, 3, padding=1),
            nn.ReLU(),
        )

        self.up1 = nn.ConvTranspose2d(128, 64, 2, stride=2)
```

```
        self.dec1 = nn.Sequential(
            nn.Conv2d(128, 64, 3, padding=1),
            nn.ReLU(),
            nn.Conv2d(64, 64, 3, padding=1),
            nn.ReLU(),
        )

        self.conv_last = nn.Conv2d(64, 1, 1)

    def center_crop(self, layer, target_h, target_w):
        _, _, h, w = layer.size()
        diff_y = h - target_h
        diff_x = w - target_w
        cropped = layer[
            :,
            :,
            diff_y // 2 : diff_y // 2 + target_h,
            diff_x // 2 : diff_x // 2 + target_w,
        ]
        return cropped

    def forward(self, x):
        # Encoder
        enc1 = self.enc1(x)
        pool1 = self.pool1(enc1)

        enc2 = self.enc2(pool1)
        pool2 = self.pool2(enc2)

        enc3 = self.enc3(pool2)
        pool3 = self.pool3(enc3)

        # Bottleneck
        bottleneck = self.bottleneck(pool3)

        # Decoder
        up3 = self.up3(bottleneck)
        enc3_cropped = self.center_crop(enc3, up3.size(2), up3.size(3))
        cat3 = torch.cat([up3, enc3_cropped], dim=1)
        dec3 = self.dec3(cat3)

        up2 = self.up2(dec3)
        enc2_cropped = self.center_crop(enc2, up2.size(2), up2.size(3))
        cat2 = torch.cat([up2, enc2_cropped], dim=1)
        dec2 = self.dec2(cat2)

        up1 = self.up1(dec2)
        enc1_cropped = self.center_crop(enc1, up1.size(2), up1.size(3))
        cat1 = torch.cat([up1, enc1_cropped], dim=1)
        dec1 = self.dec1(cat1)

        out = self.conv_last(dec1)
        out = torch.sigmoid(out)
        return out

# Instantiate model, loss function, optimizer
model = UNet().to(device)
criterion = nn.MSELoss()
optimizer = optim.Adam(model.parameters(), lr=1e-4)

# Training loop
num_epochs = 5
for epoch in range(num_epochs):
    model.train()
    running_loss = 0.0
    for inputs, targets in train_loader:
        inputs = inputs.to(device)
        targets = targets.to(device)

        optimizer.zero_grad()
        outputs = model(inputs)
        loss = criterion(outputs, targets)
        loss.backward()
        optimizer.step()

        running_loss += loss.item() * inputs.size(0)
    epoch_loss = running_loss / len(train_loader.dataset)
    print(f"Epoch {epoch+1}/{num_epochs}, Training Loss: {epoch_loss:.6f}")

    # Validation
    model.eval()
    val_loss = 0.0
    with torch.no_grad():
        for inputs, targets in val_loader:
            inputs = inputs.to(device)
            targets = targets.to(device)

            outputs = model(inputs)
            loss = criterion(outputs, targets)
            val_loss += loss.item() * inputs.size(0)
    val_loss /= len(val_loader.dataset)
```

```
1836      print(f"Epoch {epoch+1}/{num_epochs}, Validation Loss: {val_loss:.6f}")
1837
1838
1839   # Compute RMSE on validation set
       def compute_rmse(model, loader):
1840       model.eval()
           mse = 0.0
1841       num_pixels = 0
           with torch.no_grad():
1842           for inputs, targets in loader:
                   inputs = inputs.to(device)
1843               targets = targets.to(device)
                   outputs = model(inputs)
1844               mse += F.mse_loss(outputs, targets, reduction="sum").item()
                   num_pixels += targets.numel()
1845       rmse = math.sqrt(mse / num_pixels)
           return rmse
1846
1847
       rmse = compute_rmse(model, val_loader)
1848   print(f"Validation RMSE: {rmse}")
1849
       # Predict on test images
1850   model.eval()
       test_images = sorted(glob.glob("./test/*.png"))
1851   ids = []
       vals = []
1852   transform = transforms.Compose([transforms.ToTensor()])
       for img_path in test_images:
1853       img = Image.open(img_path).convert("L")
           img_id = os.path.basename(img_path).split(".")[0]
1854       input_tensor = transform(img).unsqueeze(0).to(device)
           # Pad the input tensor to next multiple of 8
1855       _, _, h, w = input_tensor.size()
           new_h = ((h - 1) // 8 + 1) * 8
1856       new_w = ((w - 1) // 8 + 1) * 8
           pad_h = new_h - h
1857       pad_w = new_w - w
           padding = (0, pad_w, 0, pad_h) # left, right, top, bottom
1858       input_tensor = F.pad(input_tensor, padding)
           with torch.no_grad():
1859           output = model(input_tensor)
           # Unpad the output tensor
1860       output = output[:, :, :h, :w]
           output = output.squeeze().cpu().numpy()
1861       # Ensure values are between 0 and 1
           output = np.clip(output, 0, 1)
1862       h, w = output.shape
           for i in range(h):
1863           for j in range(w):
                   ids.append(f"{img_id}_{i+1}_{j+1}")
1864               vals.append(output[i, j])
1865
       # Create submission dataframe
1866   submission = pd.DataFrame({"id": ids, "value": vals})
1867
       # Ensure the submission matches the sample submission format
1868   sample_submission = pd.read_csv("./sampleSubmission.csv")
       submission = submission.merge(sample_submission[["id"]], on="id", how="right")
1869   submission = submission.fillna(0)
       submission = submission[["id", "value"]]
1870
       # Save submission
1871   submission.to_csv("./submission.csv", index=False)
       print("Submission file saved to ./submission.csv")
1872
       # Print validation RMSE
1873   print(f"Final Validation RMSE: {rmse}")
```

### C.3.3    ML-AGENT TRAJECTORY ON denoising-dirty-documents

Partial trajectory of ML-Agent on denoising-dirty-documents

```
==================== Step 7 ====================
Reflection: The modified training script executed successfully, and the validation RMSE has improved to approximately 0.0838. This
    is a slight improvement compared to the previous RMSE of 0.0884. However, it still does not meet the target of reducing the
    RMSE below 0.01. The training loss decreased consistently over the epochs, indicating that the model is learning effectively.
    Further modifications are necessary to achieve the desired performance.
Research Plan and Status: 1. Understand the 'train.py' script: Completed. The model architecture, training process, hyperparameters,
    and evaluation methods have been identified.
2. Identify potential modifications: Completed. The first modification (changing the learning rate) has been implemented and
    evaluated. The second modification (increasing the number of epochs) has been implemented and evaluated. The third
    modification (changing the optimizer) has been implemented and evaluated.
3. Implement modifications: The optimizer has been modified in 'train_modified_optimizer.py', resulting in a validation RMSE of
    approximately 0.0838.
4. Iterate: I will now explore further modifications, such as adjusting the batch size or experimenting with different learning
    rates, to further reduce the RMSE.

Current Status: Step 3 is completed. I will now proceed to step 4 and explore further modifications to the training script.
Fact Check: 1. The validation RMSE after changing the optimizer is approximately 0.0838. (Confirmed)
```

```
Thought: I will experiment with adjusting the learning rate from 1e-4 to 1e-5, as a smaller learning rate may help the model
    converge more effectively and potentially lead to a lower RMSE.

Action: Edit Script (AI)
Action Input: {
    "script_name": "train_modified_optimizer.py",
    "edit_instruction": "Change the learning rate from 1e-4 to 1e-5.",
    "save_name": "train_modified_optimizer_lr.py"
}
Observation: The edited file is saved to train_modified_optimizer_lr.py. Here is the diff, please check if the edit is correct and
    desirable:

---
+++
@@ -223,7 +223,7 @@
 # Instantiate model, loss function, optimizer
 model = UNet().to(device)
 criterion = nn.MSELoss()
-optimizer = optim.RMSprop(model.parameters(), lr=1e-4)
+optimizer = optim.RMSprop(model.parameters(), lr=1e-5)

 # Training loop
 num_epochs = 10

=================== Step 8 ===================
Reflection: The modification to change the learning rate from 1e-4 to 1e-5 has been successfully made. This change aims to allow the
    model to converge more effectively, potentially leading to a reduction in the RMSE. The next step is to execute the modified
    training script to evaluate its impact on the RMSE.
Research Plan and Status: 1. Understand the 'train.py' script: Completed. The model architecture, training process, hyperparameters,
    and evaluation methods have been identified.
2. Identify potential modifications: Completed. The first modification (changing the learning rate) has been implemented and
    evaluated. The second modification (increasing the number of epochs) has been implemented and evaluated. The third
    modification (changing the optimizer) has been implemented and evaluated. The fourth modification (changing the learning rate)
    has been implemented.
3. Implement modifications: The learning rate has been modified in 'train_modified_optimizer_lr.py'.
4. Iterate: Execute the modified script to evaluate the impact on RMSE.

Current Status: Step 3 is completed. I will now proceed to step 4 and execute the modified training script.
Fact Check: 1. The learning rate has been changed from 1e-4 to 1e-5. (Confirmed)
Thought: I will execute the modified training script 'train_modified_optimizer_lr.py' to see if changing the learning rate leads to
    an improvement in the RMSE on the validation set.

Action: Execute Script
Action Input: {
    "script_name": "train_modified_optimizer_lr.py"
}
Observation: Epoch 1/10, Training Loss: 0.208905
Epoch 1/10, Validation Loss: 0.202602
Epoch 2/10, Training Loss: 0.168855
Epoch 2/10, Validation Loss: 0.110859
Epoch 3/10, Training Loss: 0.107715
Epoch 3/10, Validation Loss: 0.101807
Epoch 4/10, Training Loss: 0.102095
Epoch 4/10, Validation Loss: 0.099179
Epoch 5/10, Training Loss: 0.096579
Epoch 5/10, Validation Loss: 0.096960
Epoch 6/10, Training Loss: 0.097088
Epoch 6/10, Validation Loss: 0.093844
Epoch 7/10, Training Loss: 0.091358
Epoch 7/10, Validation Loss: 0.090865
Epoch 8/10, Training Loss: 0.090154
Epoch 8/10, Validation Loss: 0.086134
Epoch 9/10, Training Loss: 0.084577
Epoch 9/10, Validation Loss: 0.080215
Epoch 10/10, Training Loss: 0.075586
Epoch 10/10, Validation Loss: 0.068712
Validation RMSE: 0.26212960276846825
Submission file saved to ./submission.csv
Final Validation RMSE: 0.26212960276846825

=================== Step 9 ===================
Reflection: The modified training script executed successfully, but the validation RMSE has increased to approximately 0.2621, which
    is a significant regression compared to the previous RMSE of 0.0838. This indicates that reducing the learning rate to 1e-5
    may have hindered the model's ability to learn effectively within the given epochs. It suggests that the learning rate might
    be too low for this specific model and dataset, leading to slower convergence or getting stuck in local minima.
Research Plan and Status: 1. Understand the 'train.py' script: Completed. The model architecture, training process, hyperparameters,
    and evaluation methods have been identified.
2. Identify potential modifications: Completed. The first modification (changing the learning rate) has been implemented and
    evaluated. The second modification (increasing the number of epochs) has been implemented and evaluated. The third
    modification (changing the optimizer) has been implemented and evaluated. The fourth modification (changing the learning rate)
    has been implemented and evaluated.
3. Implement modifications: The learning rate has been modified in 'train_modified_optimizer_lr.py', resulting in a validation RMSE
    of approximately 0.2621.
4. Iterate: I will revert the learning rate back to 1e-4 and explore other modifications, such as adjusting the batch size or
    experimenting with different optimizers, to further reduce the RMSE.

Current Status: Step 3 is completed. I will now proceed to step 4 and revert the learning rate back to 1e-4 in the training script.
Fact Check: 1. The validation RMSE after changing the learning rate to 1e-5 is approximately 0.2621. (Confirmed)
Thought: I will revert the learning rate back to 1e-4 in the script and execute it again to see if returning to the previous
    learning rate improves the RMSE.

Action: Edit Script (AI)
```

```
Action Input: {
    "script_name": "train_modified_optimizer_lr.py",
    "edit_instruction": "Change the learning rate back from 1e-5 to 1e-4.",
    "save_name": "train_reverted_lr.py"
}
Observation: The edited file is saved to train_reverted_lr.py. Here is the diff, please check if the edit is correct and desirable:

---
+++
@@ -223,7 +223,7 @@
 # Instantiate model, loss function, optimizer
 model = UNet().to(device)
 criterion = nn.MSELoss()
-optimizer = optim.RMSprop(model.parameters(), lr=1e-5)
+optimizer = optim.RMSprop(model.parameters(), lr=1e-4)

 # Training loop
 num_epochs = 10
```

### C.3.4  ANALYSIS

As shown in C.3.3, we see that ML-Agent automatically initiates backtracking mechanisms when performance metrics remain unimproved by proposed modifications. It tries to use a smaller learning rate in step 7 but finds the RMSE increases after executing the code in step 8. Thus, it tries to revert the learning rate back out its initial value in step 9. We also show some step examples on different tasks in this section, which shows that ML-Agent can generate diverse action sequences through comprehensive reasoning. As a comparison, other models (such as GPT-4o) sometimes fail to understand the requirement and submit the answer after only one modification(sometimes even no modification).

Some step examples on different tasks

```
=================== Step 13 ===================
Reflection: The observation indicates that the validation MAE is currently at 18678, which is significantly above the target of 5000.
    This suggests that the feature engineering steps taken so far, including the logarithmic transformations, have not
    sufficiently improved the model's performance. Further modifications are needed to reduce the MAE.
Research Plan and Status: 1. Understand the train.py script: Completed. The current features and model have been identified.
2. Feature Engineering: Logarithmic transformations were applied to skewed numerical features, but the validation MAE remains high
    at 18678.
3. Model Evaluation: The model's performance needs further improvement, as the validation MAE is still above the target.
4. Iterate: I will explore additional feature engineering techniques, such as creating new interaction features or transforming
    existing features further.

Current Status: The validation MAE is still above the target, and I will now consider creating additional interaction features or
    transforming existing features to see if that helps reduce the MAE.
Fact Check: 1. The validation MAE is confirmed to be 18678, as confirmed by the observation.
Thought: I will edit the 'train.py' script to create additional interaction features between some of the existing features, which
    may help improve the model's performance.

Action: Edit Script (AI)
Action Input: {
    "script_name": "train.py",
    "edit_instruction": "Create new interaction features by multiplying 'OverallQual' with 'GrLivArea' and 'TotalBsmtSF' with '
        GarageCars'. Add these new features to the dataset.",
    "save_name": "train.py"
}
Observation: The edited file is saved to train.py. Here is the diff, please check if the edit is correct and desirable:

---
+++
@@ -30,6 +30,10 @@
    'YearBuilt_TotalBsmtSF', 'YearBuilt_GrLivArea',
    'TotalBsmtSF_GrLivArea'
 ]
+
+# Add new interaction features
+interaction_features.append('OverallQual_GrLivArea')
+interaction_features.append('TotalBsmtSF_GarageCars')

 for feature in interaction_features:
     cols = feature.split('_')
```

Some step examples on different tasks

```
=================== Step 5 ===================
Reflection: The modified training script with data augmentation has been executed successfully. However, the validation RMSE is
    0.36237191602303853, which is still significantly above the target of 0.01. This indicates that while data augmentation may
    have improved the training process, it was not sufficient to achieve the desired performance.

Next, I will consider another modification to further reduce the RMSE. Given the potential areas for improvement identified earlier,
    I will explore increasing the model complexity by adding more layers to the U-Net architecture. This could help the model
    learn more complex patterns in the data.
Research Plan and Status: 1. Understand the 'train.py' script: Completed. The script uses a modified U-Net for image denoising with
    a fixed learning rate and no data augmentation.
2. Identify potential modifications: Identified several potential modifications, including learning rate scheduling, data
    augmentation, model complexity, regularization, and early stopping.
```

```
3. Implement the modification: Completed. A learning rate scheduler has been added to the training loop.
4. Evaluate results: Completed. The validation RMSE after the learning rate scheduler modification is 0.2463417127589059.
5. Next modification: Completed. Data augmentation has been added to the training data transformation pipeline.
6. Evaluate results again: Completed. The validation RMSE after data augmentation is 0.36237191602303853.
7. Next modification: I will increase the model complexity by adding more layers to the U-Net architecture.

Current Status:
- Step 1 is completed.
- Step 2 is completed.
- Step 3 is completed.
- Step 4 is completed.
- Step 5 is completed.
- Step 6 is completed.
- Step 7 is in progress as I will now modify the script to increase model complexity.
Fact Check: 1. The final validation RMSE after the data augmentation modification is 0.36237191602303853. (Confirmed)
Thought: I will proceed to modify the `train.py` script to increase the complexity of the U-Net model by adding additional
      convolutional layers and possibly increasing the number of filters in existing layers. This should help the model capture
      more complex features and potentially reduce the RMSE.

Action: Edit Script (AI)
Action Input: {
    "script_name": "train.py",
    "edit_instruction": "Increase the complexity of the U-Net model by adding two additional convolutional layers with 64 filters
        each after the existing layers. Also, increase the number of filters in the first layer to 32.",
    "save_name": "train_with_complex_model.py"
}
Observation: The edited file is saved to train_with_complex_model.py. Here is the diff, please check if the edit is correct and
      desirable:

---
+++
@@ -115,63 +115,87 @@
        super(UNet, self).__init__()
        # Encoder
        self.enc1 = nn.Sequential(
- nn.Conv2d(1, 64, 3, padding=1),
- nn.ReLU(),
- nn.Conv2d(64, 64, 3, padding=1),
+ nn.Conv2d(1, 32, 3, padding=1),
+ nn.ReLU(),
+ nn.Conv2d(32, 32, 3, padding=1),
+ nn.ReLU(),
+ nn.Conv2d(32, 32, 3, padding=1),
+ nn.ReLU(),
+ nn.Conv2d(32, 32, 3, padding=1),
        nn.ReLU(),
        )
        self.pool1 = nn.MaxPool2d(2, 2)

        self.enc2 = nn.Sequential(
+ nn.Conv2d(32, 64, 3, padding=1),
+ nn.ReLU(),
+ nn.Conv2d(64, 64, 3, padding=1),
+ nn.ReLU(),
+ nn.Conv2d(64, 64, 3, padding=1),
+ nn.ReLU(),
+ nn.Conv2d(64, 64, 3, padding=1),
+ nn.ReLU(),
+ )
+ self.pool2 = nn.MaxPool2d(2, 2)
+
+ self.enc3 = nn.Sequential(
        nn.Conv2d(64, 128, 3, padding=1),
        nn.ReLU(),
        nn.Conv2d(128, 128, 3, padding=1),
        nn.ReLU(),
- )
- self.pool2 = nn.MaxPool2d(2, 2)
-
- self.enc3 = nn.Sequential(
- nn.Conv2d(128, 256, 3, padding=1),
- nn.ReLU(),
- nn.Conv2d(256, 256, 3, padding=1),
+ nn.Conv2d(128, 128, 3, padding=1),
+ nn.ReLU(),
+ nn.Conv2d(128, 128, 3, padding=1),
        nn.ReLU(),
        )
        self.pool3 = nn.MaxPool2d(2, 2)

        # Bottleneck
        self.bottleneck = nn.Sequential(
- nn.Conv2d(256, 512, 3, padding=1),
- nn.ReLU(),
- nn.Conv2d(512, 512, 3, padding=1),
+ nn.Conv2d(128, 256, 3, padding=1),
+ nn.ReLU(),
+ nn.Conv2d(256, 256, 3, padding=1),
        nn.ReLU(),
        )

        # Decoder
```

```
- self.up3 = nn.ConvTranspose2d(512, 256, 2, stride=2)
+ self.up3 = nn.ConvTranspose2d(256, 128, 2, stride=2)
        self.dec3 = nn.Sequential(
- nn.Conv2d(512, 256, 3, padding=1),
- nn.ReLU(),
- nn.Conv2d(256, 256, 3, padding=1),
- nn.ReLU(),
- )
-
- self.up2 = nn.ConvTranspose2d(256, 128, 2, stride=2)
+ nn.Conv2d(256, 128, 3, padding=1),
+ nn.ReLU(),
+ nn.Conv2d(128, 128, 3, padding=1),
+ nn.ReLU(),
+ nn.Conv2d(128, 128, 3, padding=1),
+ nn.ReLU(),
+ nn.Conv2d(128, 128, 3, padding=1),
+ nn.ReLU(),
+ )
+
+ self.up2 = nn.ConvTranspose2d(128, 64, 2, stride=2)
        self.dec2 = nn.Sequential(
- nn.Conv2d(256, 128, 3, padding=1),
- nn.ReLU(),
- nn.Conv2d(128, 128, 3, padding=1),
- nn.ReLU(),
- )
-
- self.up1 = nn.ConvTranspose2d(128, 64, 2, stride=2)
+ nn.Conv2d(128, 64, 3, padding=1),
+ nn.ReLU(),
+ nn.Conv2d(64, 64, 3, padding=1),
+ nn.ReLU(),
+ nn.Conv2d(64, 64, 3, padding=1),
+ nn.ReLU(),
+ nn.Conv2d(64, 64, 3, padding=1),
+ nn.ReLU(),
+ )
+
+ self.up1 = nn.ConvTranspose2d(64, 32, 2, stride=2)
        self.dec1 = nn.Sequential(
- nn.Conv2d(128, 64, 3, padding=1),
- nn.ReLU(),
- nn.Conv2d(64, 64, 3, padding=1),
- nn.ReLU(),
- )
-
- self.conv_last = nn.Conv2d(64, 1, 1)
+ nn.Conv2d(64, 32, 3, padding=1),
+ nn.ReLU(),
+ nn.Conv2d(32, 32, 3, padding=1),
+ nn.ReLU(),
+ nn.Conv2d(32, 32, 3, padding=1),
+ nn.ReLU(),
+ nn.Conv2d(32, 32, 3, padding=1),
+ nn.ReLU(),
+ )
+
+ self.conv_last = nn.Conv2d(32, 1, 1)

    def center_crop(self, layer, target_h, target_w):
        _, _, h, w = layer.size()
@@ -323,4 +347,4 @@
 print("Submission file saved to ./submission.csv")

 # Print validation RMSE
-print(f"Final Validation RMSE: {rmse}")
+print(f"Final Validation RMSE: {rmse}")
```

## Some step examples on different tasks

```
==================== Step 7 ====================
Reflection: The modified training script has been executed successfully, but the quadratic weighted kappa score achieved is 0.7418,
    which is slightly lower than the previous score of 0.7431. This indicates that the changes made to the learning rate and the
    number of boosting stages did not lead to an improvement in performance. Further modifications are necessary to enhance the
    model's performance.
Research Plan and Status: 1. Understand the 'train.py' script: Completed. The model architecture, training process, and
    hyperparameters have been identified.
2. Experiment with modifications: Completed. The learning rate was changed to 0.05 and the number of boosting stages increased to
    200, but the performance decreased slightly.
3. Evaluate performance: Completed. The modified training script was executed, resulting in a validation quadratic weighted kappa
    score of 0.7418.
4. Iterate: I will now explore further modifications to improve the model's performance.

Current Status:
- Step 4 is in progress. I will consider additional modifications, such as trying different feature extraction techniques or
    adjusting other hyperparameters, to enhance the model's performance.

Next Step: I will try to modify the feature extraction process again by using a different vectorization method, such as using a more
    advanced model like Sentence-BERT, which may capture more contextual information from the essays. However, since we cannot
    directly modify the vectorization method in the current setup, I will consider using a simpler approach like averaging the TF-
    IDF scores across different n-grams.
```

```
Fact Check: 1. The validation quadratic weighted kappa score achieved after the third modification is 0.7418. (Confirmed by
    observation)
Thought: I will proceed to modify the 'train.py' script to implement a different feature extraction method that averages the TF-IDF
    scores across different n-grams. This change aims to improve the model's performance.

Action: Edit Script (AI)
Action Input: {
    "script_name": "train.py",
    "edit_instruction": "Modify the feature extraction process to average the TF-IDF scores across different n-grams (bigrams and
        trigrams) instead of using the default TF-IDF vectorization.",
    "save_name": "train_feature_extraction_modified.py"
}
Observation: The edited file is saved to train_feature_extraction_modified.py. Here is the diff, please check if the edit is correct
    and desirable:

---
+++
@@ -10,10 +10,22 @@
 train_df = pd.read_csv("./train.csv")
 test_df = pd.read_csv("./test.csv")

-# Prepare the features using TF-IDF vectorization
-tfidf = TfidfVectorizer(max_features=10000)
-X = tfidf.fit_transform(train_df["full_text"])
-X_test = tfidf.transform(test_df["full_text"])
+# Prepare the features using TF-IDF vectorization with bigrams and trigrams
+tfidf_unigrams = TfidfVectorizer(max_features=5000, ngram_range=(1, 1))
+tfidf_bigrams = TfidfVectorizer(max_features=5000, ngram_range=(2, 2))
+tfidf_trigrams = TfidfVectorizer(max_features=5000, ngram_range=(3, 3))
+
+X_unigrams = tfidf_unigrams.fit_transform(train_df["full_text"])
+X_bigrams = tfidf_bigrams.fit_transform(train_df["full_text"])
+X_trigrams = tfidf_trigrams.fit_transform(train_df["full_text"])
+
+X_test_unigrams = tfidf_unigrams.transform(test_df["full_text"])
+X_test_bigrams = tfidf_bigrams.transform(test_df["full_text"])
+X_test_trigrams = tfidf_trigrams.transform(test_df["full_text"])
+
+# Average the TF-IDF scores across unigrams, bigrams, and trigrams
+X = (X_unigrams + X_bigrams + X_trigrams) / 3
+X_test = (X_test_unigrams + X_test_bigrams + X_test_trigrams) / 3
 y = train_df["score"]

 # Split the data into training and validation sets
@@ -52,4 +64,4 @@
 submission = pd.DataFrame({"essay_id": test_df["essay_id"], "score": test_pred_round})

 # Save the submission file
-submission.to_csv("./submission.csv", index=False)
+submission.to_csv("./submission.csv", index=False)
```

## Some step examples on different tasks

```
==================== Step 1 ====================
Reflection: The observation provides a comprehensive overview of the 'train.py' script, detailing its structure, the model used, and
    the training process. It highlights the key components such as data reading, label mapping, data splitting, text
    vectorization, model training, prediction, and evaluation. However, it also notes that there is no preprocessing of text data
    before vectorization, no feature selection or dimensionality reduction, and no cross-validation or hyperparameter tuning.
    This indicates potential areas for improvement.
Research Plan and Status: 1. Understand the 'train.py' script: Completed. The script uses a 'LogisticRegression' model and lacks
    preprocessing and tuning steps.
2. Identify potential modifications:
    - **First Modification**: Implement text preprocessing (tokenization, stopword removal, stemming/lemmatization) before
        vectorization to improve feature representation.
3. Implement the modification: Modify the script to include text preprocessing.
4. Iterate: After implementing the first modification, run the training process and evaluate the impact on loss. If the loss does
    not improve, consider further modifications.

Current Status: The first modification has been identified and will be implemented next.
Fact Check: 1. The script uses a 'LogisticRegression' classifier for the text classification task. (Confirmed)
2. The script lacks preprocessing of text data before vectorization. (Confirmed)
3. The script does not perform feature selection or dimensionality reduction. (Confirmed)
4. The script does not include cross-validation or hyperparameter tuning. (Confirmed)
Thought: I will proceed to modify the 'train.py' script to include text preprocessing steps before the vectorization process. This
    should help improve the model's performance by providing better features.

Action: Edit Script (AI)
Action Input: {
    "script_name": "train.py",
    "edit_instruction": "Add text preprocessing steps before vectorization. Use NLTK for tokenization, stopword removal, and
        stemming. Specifically, import nltk and use nltk's tokenization, stopwords, and PorterStemmer. Apply these steps to the
        text data before vectorization.",
    "save_name": "train.py"
}
Observation: The edited file is saved to train.py. Here is the diff, please check if the edit is correct and desirable:

---
+++
@@ -5,6 +5,14 @@
 from sklearn.linear_model import LogisticRegression
 from sklearn.metrics import log_loss
 import os
```

```
+import nltk
+from nltk.tokenize import word_tokenize
+from nltk.corpus import stopwords
+from nltk.stem import PorterStemmer
+
+# Download NLTK data files (only need to run once)
+nltk.download('punkt')
+nltk.download('stopwords')

 # Read the data
 train = pd.read_csv("./train.csv")
@@ -13,6 +21,22 @@
 # Preprocess labels
 author_mapping = {"EAP": 0, "HPL": 1, "MWS": 2}
 train["author_num"] = train["author"].map(author_mapping)
+
+# Initialize NLTK components
+stop_words = set(stopwords.words('english'))
+stemmer = PorterStemmer()
+
+# Define a function for text preprocessing
+def preprocess_text(text):
+ # Tokenize the text
+ tokens = word_tokenize(text.lower())
+ # Remove stopwords and apply stemming
+ tokens = [stemmer.stem(word) for word in tokens if word.isalpha() and word not in stop_words]
+ return ' '.join(tokens)
+
+# Apply preprocessing to the text data
+train["text"] = train["text"].apply(preprocess_text)
+test["text"] = test["text"].apply(preprocess_text)

 # Split the data into training and validation sets
 X_train, X_val, y_train, y_val = train_test_split(
@@ -53,4 +77,4 @@
 submission = submission[["id", "EAP", "HPL", "MWS"]]

 # Save submission
-submission.to_csv("./submission.csv", index=False)
+submission.to_csv("./submission.csv", index=False)
```

gpt-4o trajectory on detecting-insults-in-social-commentary

```
=================== Step 0 ===================
Reflection: The task involves understanding and modifying a machine learning script, 'train.py', to improve its performance on a
        dataset related to detecting insults in social commentary. The dataset includes training and test sets, and a sample
        submission file. The evaluation metric is AUC, and predictions should be in the range [0,1]. We are to make incremental
        modifications to the script to improve performance, keeping epochs to 10 or less and not altering the random seed.

Research Plan and Status:
1. **Understand the current state of the 'train.py' script**: Not started.
2. Identify areas in the script that could be modified to improve model performance: Not started.
3. Make one modification at a time to the script and test the effect on performance: Not started.
4. Aim to achieve a test accuracy of 99.99%: Not started.

Fact Check: None, as no progress has been made yet.

Thought: The first step is to understand the current state of the 'train.py' script to identify potential areas for improvement.
        This will involve inspecting the script to see what model and techniques are currently being used.

Action: List Files
Action Input: {
    "dir_path": "."
}
Observation: backup/
test.csv
train.csv
train.py

=================== Step 1 ===================
Reflection: The directory contains the 'train.py' script, alongside 'train.csv' and 'test.csv', which are the training and testing
        datasets, respectively. There is also a 'backup/' directory, which might contain backup files or previous versions of scripts.

Research Plan and Status:
1. **Understand the current state of the 'train.py' script**: Not started.
2. Identify areas in the script that could be modified to improve model performance: Not started.
3. Make one modification at a time to the script and test the effect on performance: Not started.
4. Aim to achieve a test accuracy of 99.99%: Not started.

Fact Check: The presence of 'train.py', 'train.csv', 'test.csv', and 'backup/' is directly confirmed by the file listing.

Thought: The next step is to inspect the 'train.py' script to understand the current implementation. This will help identify
        potential areas for improvement.

Action: Inspect Script Lines
Action Input: {
    "script_name": "train.py",
    "start_line_number": 1,
    "end_line_number": 100
}
Observation: Here are the lines (the file ends at line 122):
```

```python
import pandas as pd
import random
import torch
import numpy as np
from torch.utils.data import DataLoader, Dataset
import torch.nn as nn
import torch.optim as optim
from sklearn.feature_extraction.text import TfidfVectorizer
from sklearn.metrics import accuracy_score

# Set the random seed for reproducibility
def set_seed(seed):
    random.seed(seed)
    np.random.seed(seed)
    torch.manual_seed(seed)
    if torch.cuda.is_available():
        torch.cuda.manual_seed(seed)
        torch.backends.cudnn.deterministic = True
        torch.backends.cudnn.benchmark = False

set_seed(0)

# Device configuration
device = torch.device("cuda" if torch.cuda.is_available() else "cpu")

# Define a dataset class
class TextDataset(Dataset):
    def __init__(self, texts, targets):
        self.texts = texts
        self.targets = targets

    def __len__(self):
        return len(self.texts)

    def __getitem__(self, idx):
        return self.texts[idx], self.targets[idx]

# Define the neural network model
class SimpleNNClassifier(nn.Module):
    def __init__(self, input_dim):
        super(SimpleNNClassifier, self).__init__()
        self.fc1 = nn.Linear(input_dim, 64)
        self.fc2 = nn.Linear(64, 32)
        self.fc3 = nn.Linear(32, 1)

    def forward(self, x):
        x = torch.relu(self.fc1(x))
        x = torch.relu(self.fc2(x))
        x = torch.sigmoid(self.fc3(x))
        return x.squeeze()

# Train the model
def train_model(X_train, y_train):
    train_dataset = TextDataset(torch.FloatTensor(X_train), torch.FloatTensor(y_train))
    train_loader = DataLoader(train_dataset, batch_size=32, shuffle=True,num_workers=4)

    model = SimpleNNClassifier(input_dim=X_train.shape[1]).to(device)
    criterion = nn.BCELoss()
    optimizer = optim.SGD(model.parameters(), lr=0.1,momentum=0.9)

    for epoch in range(10):
        model.train()

        for texts, targets in train_loader:
            texts, targets = texts.to(device), targets.to(device)
            optimizer.zero_grad()
            outputs = model(texts)
            loss = criterion(outputs, targets)
            loss.backward()
            optimizer.step()

        y_test_pred = predict(model, X_test_tfidf)
        test_acc = accuracy_score(y_test, y_test_pred)
        print(f"Epoch {epoch} | Test Accuracy: {test_acc*100:.2f}%")

    return model

# Predict function
def predict(model, X):
    X_tensor = torch.FloatTensor(X).to(device)
    model.eval()
    with torch.no_grad():
        y_pred = model(X_tensor).cpu().numpy()
    return (y_pred > 0.5).astype(int)

# Main script
if __name__ == '__main__':
    train_data = pd.read_csv("train.csv")
    test_data = pd.read_csv("test.csv")

    X = train_data['Comment']
```

```
    y = train_data['Insult'].values

    X_test_comments = test_data['Comment']
    y_test = test_data['Insult'].values

    vectorizer = TfidfVectorizer(stop_words='english', max_features=500)
    X_tfidf = vectorizer.fit_transform(X).toarray()
    X_test_tfidf = vectorizer.transform(X_test_comments).toarray()

=================== Step 2 ====================
Reflection: The 'train.py' script sets up a simple neural network using PyTorch to classify comments as insults or not based on text
      data. It uses TF-IDF to vectorize the text and a simple feedforward neural network with two hidden layers. The model is
      trained over 10 epochs with a batch size of 32 using stochastic gradient descent (SGD) with a learning rate of 0.1 and
      momentum of 0.9. The script also includes a function to set random seeds for reproducibility and uses a binary cross-entropy
      loss function.

Research Plan and Status:
1. **Understand the current state of the 'train.py' script**: Completed. The script uses a simple neural network with TF-IDF
      features.
2. Identify areas in the script that could be modified to improve model performance: Not started.
3. Make one modification at a time to the script and test the effect on performance: Not started.
4. Aim to achieve a test accuracy of 99.99%: Not started.

Fact Check:
- The script uses TF-IDF vectorization and a simple neural network confirmed by inspection.
- The script trains the model for 10 epochs, confirmed by inspection.
- The learning rate is 0.1 with SGD optimizer, confirmed by inspection.

Thought: The next step is to identify potential areas in the script that can be modified to improve performance. Possible areas
      include model architecture, feature extraction, optimization parameters, and data preprocessing.

Action: Final Answer
Action Input: {
    "final_answer": "The 'train.py' script uses TF-IDF vectorization and a simple feedforward neural network with two hidden layers
          to classify comments as insults. It uses SGD with a learning rate of 0.1 and momentum of 0.9, training for 10 epochs. The
          key areas for potential improvement include model architecture, feature extraction, optimization parameters, and data
          preprocessing."
}
Observation: end
```

# D  THE USE OF LARGE LANGUAGE MODELS

In the preparation of this manuscript, Large Language Models (LLMs) were used as an assistive tool to improve grammar, clarity, and readability for some paragraphs. The authors reviewed and edited all text and take full responsibility for the final content of the paper.

