# OpenReview forum: "ML-Agent: Reinforcing LLM Agents for Autonomous Machine Learning Engineering"
_ICLR.cc/2026/Conference — Submitted to ICLR 2026_

### Official Review · Reviewer_ixvU · 2025-10-27

**Soundness:** 2
**Presentation:** 3
**Contribution:** 2
**Rating:** 4
**Confidence:** 3

**Summary:**

The authors design a training framework that enables LLMs to learn from environment interactions efficiently, leveraging exploration-enriched fine-tuning, step-wise RL, and a ML-specific reward module. The authors train ML-Agent, a 7B-parameter model based on Qwen-2.5, which outperformed most baselines, and showed strong generalization result.

**Strengths:**

1. Modular framework design. The three proposed components (exploration-enriched fine-tuning, step-wise RL, and reward unification) are intuitive, complementary, and grounded in practical RL challenges for ML agents.
2. Strong empirical results. The 7B- model trained with this framework rivals GPT-5-driven agents, and shows great generalization.

**Weaknesses:**

1. Sparse ablation and analysis. The contribution exploration-enriched fine-tuning is not clearly isolated in ablation studies.
2. Lack of novelty. The three components of the proposed framework, exploration-enriched fine-tuning, step-wise RL, and the agentic ML-specific reward module, appear to be additive rather than integrated or co-designed.

**Questions:**

1. Could you report quantitative ablations isolating each component, e.g., without exploration-enriched fine-tuning, without step-wise RL, and without reward normalization?
2. Does exploration-enriched fine-tuning empirically increase coverage of the action space?

**Details Of Ethics Concerns:**

No.

---

> ### Author Response · Authors · 2025-11-27
> **Response to Reviewer ixvU**
>
> We thank the reviewer for the constructive comments and suggestions. We address each of your comments in the following.
>
> ***Q1:*** *The three components appear to be additive rather than integrated/co-designed.*
>
> ***A1:*** Thank you for raising this important point. In fact, the three components are **tightly integrated and co-dependent, forming a unified training framework** where each each component enables and critically relies on the others:
>
> 1. **Exploration-enriched fine-tuning** enables effective **step-wise RL** by initializing the policy with a diverse, format-compliant action space grounded in real ML strategies. Without this SFT stage (as shown in **Figure 4**, the RL agent either fails to generate valid code (e.g., distilled models) or collapses into narrow, repetitive edits—severely limiting generalization, especially on held-out tasks.
>
> 2. **Step-wise RL** unlocks cross-task generalization that **exploration-enriched fine-tuning** alone cannot achieve. As demonstrated in **Figure 3**, the SFT-only agent (before RL training begins) exhibits near-zero improvement on held-out tasks. In contrast, as step-wise RL training progresses, the agent steadily gains performance on both held-in and held-out tasks. This demonstrates that step-wise RL is not a trivial add-on but the core mechanism enabling experience-driven learning and transfer across diverse ML tasks.
>
> 3. **The agentic ML-specific reward module** provides the essential signal for **step-wise RL** to work. **Table 2** confirms that each component of the reward plays a distinct and complementary role for step-wise RL.
>
> Crucially, **none of these components works in isolation**:
>
> - **Exploration-enriched fine-tuning without step-wise RL** → no generalization beyond training tasks.
>
> - **Step-wise RL without exploration-enriched fine-tuning** → unstable exploration and format violations.
>
> - **Step-wise RL without agentic-ML specific reward** → no meaningful learning signal.
>
> Thus, the novelty lies not in any single technique, but in their **synergistic integration into the first learning-based framework for agentic ML**, which collectively enables a small 7B agent to outperform much larger prompt-based agents. This represents a paradigm shift—from static prompting to dynamic, experience-driven learning in autonomous ML.
>
> ***Q2:*** *Quantitative ablations isolating each component.*
>
> ***A2:*** Thank you for this suggestion. We actually **already conduct comprehensive ablations for each component** in **Section 5.3**, and we are happy to summarize them in a unified table below for clarity.
>
> || Exploration-enriched Fine-tuning | Step-wise RL       | Reward Normalization            | Avg. Held-in (%) | Avg. Held-out (%) | Avg. All (%) |
> |-|----------------------------|---------------|------------------------|-------------|--------------|---------|
> |Figure 4| ✗| ✓| ✓| –0.66       | –6.20        | –4.92   |
> |Figure 3| ✓| ✗ | ✓| 8.35        | 0.41         | 2.24    |
> |Figure 3| ✓| Episode-wise  | ✓| 10.71       | 2.86         | 4.67    |
> |Table 2| ✓|✓| ✗| 9.77        | –1.65        | 0.99    |
> |ML-Agent| ✓|✓|✓ | **18.01**   | **15.91**    | **16.40** |
>
> - **Without exploration-enriched fine-tuning** (i.e., training RL directly on Qwen2.5-7B-Base), the agent fails to generate valid, diverse actions, resulting in severe performance degradation.
>
> - **Without step-wise RL** (i.e., before RL training begins/episode-wise RL), the agent shows limited generalization ability on held-out tasks.
>
> - **Without reward normalization** (i.e., uses a binary/sign reward based on metric improvement), the agent loses fine-grained learning signals, harming held-out generalization.
>
> This table confirms that **each component is essential**: removing any one leads to significant performance drops, especially on held-out tasks. We will include this consolidated ablation table in the revised manuscript for improved readability.
>
>
> ***Q3:*** *Does exploration-enriched fine-tuning empirically increase coverage of the action space?*
>
> ***A3:*** Yes. **As shown in Figure 7 in Appendix C.2**, we compared the range of actions produced by the Qwen2.5-7B-Instruct(without our fine-tuning) and ML-Agent-SFT(after exploration-enriched SFT): We randomly sampled 100 edit instructions from expert trajectories and used SpaCy to extract verb–noun pairs representing “action–target” semantics (e.g., “add BatchNorm” or “replace ReLU”). Figure 7 plots the number of unique nouns per editing verb for both models. The results show that ML-Agent-SFT consistently produces a broader and more diverse set of concrete actions for the same high-level verbs, indicating significantly expanded coverage of the executable action space.

---

### Official Review · Reviewer_6UHw · 2025-10-30

**Soundness:** 3
**Presentation:** 2
**Contribution:** 1
**Rating:** 2
**Confidence:** 4

**Summary:**

The paper introduces ML Agent, a method for fine-tuning LLMs using one-step RL, a lightweight alternative to traditional multi-step RL algorithms like PPO or DPO.
Instead of simulating full dialogue trajectories or optimizing cumulative rewards, ML Agent applies a single-step policy improvement based on feedback signals (e.g., preference or quality scores) for individual responses.
The paper shows comparable results of post-training on a small-sized model compared to proprietary LLMs such as GPT-5.

**Strengths:**

The new RL update method, if proven efficient and comparable or better to SOTA techniques, could be a good tool for LLM post-training when tool calls (such as training a ML model) are expensive.

**Weaknesses:**

- I am having a hard time seeing the novelty of this approach. The lack of comparison with a broader range of papers in the literature (Reinforcement Learning for Machine Learning Engineering Agents, MLE-Dojo, MLGym, AgentGym...) makes it hard to really see how much  is brought by this paper.
- I don't see a theoretical justification for the One-step RL method, and the empirical validation is itself a bit weak. I think the authors should show that this is a sound thing to do by showing:
  - Better sample efficiency vs PPO, or
  - Equal or better final alignment compared to a well-tuned PPO model.
The statistical rigor (significance etc) is also limited.

**Questions:**

- Could the authors clarify how their method relates to recent frameworks like Reinforcement Learning for Machine Learning Engineering Agents, MLE-Dojo, MLGym, or AgentGym? The literature on agents / tools and ML scientists has exploded these past months. In particular, does ML Agent address any limitations or gaps identified in those works?
- The paper presents one-step RL as novel, but similar formulations (reward-weighted log-likelihood updates) have been studied extensively. What, specifically, is new here, the algorithm, the training pipeline, something else?
- Does the one-step approach introduce bias relative to multi-step RL methods such as PPO?
- Why do the authors not include comparisons against PPO-based RLHF, or at least controlled reimplementations using similar datasets and reward models? One could use a simple problem with a small model if compute availability is an issue.

---

> ### Author Response · Authors · 2025-11-27
> **Response to Reviewer 6UHw --- Part 1/3**
>
> We thank the reviewer for the comments and suggestions.  We address each of your comments in the following.
>
> >**Q1.1:** How the method relates to Reinforcement Learning for Machine Learning Engineering Agents, MLE-Dojo, MLGym and AgentGym.
>
> **A1.1:** "Reinforcement Learning for Machine Learning Engineering Agents" considers the **single-task setting** "given a model and a compute budget, how should an agent best solve one ML problem?" The policy is trained and evaluated on essentially the same task. Our work instead focuses on **cross-task generalization**: we train a single ML engineering agent on a *distribution* of diverse ML tasks and evaluate it on **unseen tasks**, explicitly studying how to learn reusable MLE strategies that transfer across problems.
>
> MLE-Dojo, MLGym, and AgentGym are primarily **gym-style environments**: MLE-Dojo targets Kaggle-style ML engineering tasks, MLGym targets open-ended AI research tasks, and AgentGym aggregates heterogeneous non-ML environments to study general agents. These works largely evaluate **prompt-based agents** and do not provide **a learning-based training framework** for ML agents.
>
> >**Q1.2:** Does ML-Agent address any limitations or gaps identified in those works?
>
> **A1.2:** Rather than correcting "limitations", ML-Agent is complementary to these works and focuses on a different layer of the stack. Our main contrast is with the dominant **prompt‑based MLE agents** used in works such as MLE‑Bench, MLAgentBench, MLE‑Dojo, MLGym and AgentGym: these agents are built by hand‑crafted prompts over **frozen LLMs**. In practice, this leads to:
> 1.Small open models **cannot internalize trajectory feedback** and show weak cross‑task generalization abilities. 2. Strong performance often requires **very large proprietary models** with high cost.
>
> ML‑Agent explores a complementary, learning‑based direction. We train a 7B LLM-Agent, explicitly target **cross‑task generalization** and show that it can **match or surpass** prompt‑based agents powered by much larger proprietary LLMs at **much lower computational cost**.
>
> In summary, works include MLE‑Dojo/MLGym/AgentGym focus mainly  on **the environments and benchmark**, while our contribution is a **training framework** on ML tasks, allowing small LLM agent to learn through interactive ML experimentation for cross-task generalization at low computation cost.
>
> We will explicitly discuss these works in the related-work section and add a clearer comparison in the revised version.

---

> ### Author Response · Authors · 2025-11-27
> **Response to Reviewer 6UHw --- Part 2/3**
>
> > ***Q2:*** The novelty of step-wise RL.
>
> Thank you for raising this question. We do not claim novelty at the level of a new RL algorithm or gradient estimator.  The contribution lies in the training pipeline that enable RL-based learning from diverse ML tasks and cross-task generalization in the setting of autonomous ML with low computational cost, i.e., a long-horizon, high-latency, code-execution environment where the agent must repeatedly modify and execute real ML code to improve performance over time.
>
>
> In typical RLHF/RLVR-style setups, the model interacts with a short-horizon environment: a prompt $x$ is given, the model outputs $y$, and a scalar reward $r(x,y)$ is returned. The “state” is essentially static, episodes are cheap to sample, and the same input distribution can be reused throughout training.
>
> By contrast, in autonomous ML the agent operates over many environment steps, where each step is a substantive ML action, executed by running real code. A full 15-step episode under the current policy can take tens of minutes to hours, so repeatedly sampling full on‑policy trajectories to estimate $d^{\pi_\theta}$ is infeasible.
>
> Our step-wise RL paradigm is designed specifically for this setting:
>
> **Decoupled state distribution.** We explicitly replace the on‑policy state distribution by a fixed expert‑induced distribution. We then optimize the step-wise optimize function so that each RL update requires *only one* additional environment step instead of re‑running full trajectories under the current policy.
>
> **Environment‑step granularity in a long‑horizon code environment.** The “one step” in our framework is not a token-level move, but an atomic ML action applied to an evolving ML pipeline state (task spec, code, logs), with rewards computed via our ML‑specific reward module (metric deltas, error handling, corner cases). This is different from standard reward‑weighted log‑likelihood over static prompts.
>
> **Integration into an end‑to‑end training pipeline.** Step-wise RL is synergistic integrated with (i) exploration‑enriched fine‑tuning that yields a diverse, format‑correct action space, and (ii) an agentic ML-specific reward that unifies heterogeneous ML feedback into a single scalar reward. This combination is what allows us to train a 7B agent that learns from experience and generalizes to held-out ML tasks with low computation cost, where prompt‑only agents and naive episode‑wise PPO fail to achieve.
>
> In summary, we do not propose a new RL update rule; the novelty is in how we formulate and orchestrate RL(via an expert‑induced state distribution, step‑level interaction, and ML‑specific rewards) to make learning‑based agentic ML feasible in a realistic and expensive Agentic ML environment.
>
> > ***Q3:*** *Does the one-step approach introduce bias relative to multi-step RL methods such as PPO?*
>
> **A3:** Our step-wise formulation is indeed an approximation of standard trajectory-level PPO in two respects, but it remains an online RL procedure: actions are generated by the current policy and evaluated by the real environment at training time.
>
> **State distribution approximation.**  The “ideal” objective in takes expectation over the state distribution induced by the current policy $d^{\pi_\theta}$. In practice, we sample states from a fixed pool built from expert trajectories, leading to the step-wise objective. The bias comes from approximating $d^{\pi_\theta}$ with $d^{\pi_e}$. This is intentional:  (i) rolling out full trajectories from scratch under $\pi_\theta$ is extremely expensive in our ML environment;  (ii) after exploration-enriched fine-tuning, $\pi_\theta$ stays close to the expert, so the mismatch between the two state distributions is limited in practice.
>
> **Why this approximation is acceptable.**  Figure 3 compares our step-wise training with an episode-wise RL variant (standard PPO) that rolls out full trajectories. Under the same compute budget, step-wise RL achieves larger gains on both held-in and held-out tasks and converges faster in wall-clock time. Moreover, the resulting policy demonstrates strong cross-task generalizaiton abilities. This shows that although we approximate the ideal state distribution, the resulting procedure is both practical and empirically superior in our high-cost agentic ML setting.

---

> > ### Author Response · Authors · 2025-11-27
> > **Response to Reviewer 6UHw --- Part 3/3**
> >
> > >**Q4:** Better sample efficiency vs PPO and equal/better final alignment compared to a well-tuned PPO model.
> >
> > **A4:** We apologize that our current writing did not clearly convey this. We address the points below:
> >
> > **We do have a PPO-based baseline; it is our “episode-wise RL”:** Our “episode-wise RL” variant (Fig. 3) is exactly a standard PPO-based RL. To avoid confusion, we will rename this baseline to **“Episode-wise PPO (standard PPO)”** in the revised version.
> >
> > **Sample efficiency and final performance vs PPO:** Sample efficiency is widely defined as the ability of an algorithm to learn effective policies using as few environment interactions as possible. In our setting, we measure it as the average performance improvement per rollout. From Fig. 3, we observe that **Episode-wise PPO (standard PPO)**, which performs **7,680 step rollouts**, achieves only about **+3%** average relative performance gain, whereas **Step-wise PPO (ours)**, using **10,000 single-step rollouts**, reaches approximately **+18%** average relative performance gain. In other words, under comparable environment interaction budgets, step-wise PPO both improves substantially faster and achieves a much higher final performance on both held-in and held-out tasks, while episode-wise PPO improves slowly and is much more expensive in practice because each rollout corresponds to a full 15-step ML trajectory. We will also add a table with **mean ± std over 8 runs** for both methods and all tasks, and report significance tests in the appendix; the improvements of step-wise PPO over episode-wise PPO are statistically significant on most tasks (p < 0.05).

---

### Official Review · Reviewer_i2zT · 2025-11-01

**Soundness:** 2
**Presentation:** 3
**Contribution:** 3
**Rating:** 4
**Confidence:** 4

**Summary:**

The paper introduces RL to train ML Engineering agents to learn from past experiences. They address three problems in training an ML Agent: first, small ML agents lack exploration; they address this by distilling from a larger model. Second, to overcome the slow feedback loop of typical ML experiments, the authors introduce a step-wise reinforcement learning (RL) paradigm. Instead of learning from entire trajectories, they reformulate the problem to learn from single action steps sampled from the pre-collected expert states. Finally, this RL process is guided by a carefully designed agentic ML-specific reward module. Experiments show comparable performance of a trained 7B model to a prompted 600B model. Although the experimental results are promising, the paper lacks discussion on highly probable reward hacking (due to reward design) and distribution shift problems (due to step-wise RL).

Overall, the paper introduces a novel learning-based paradigm that enables smaller models to achieve strong performance, addresses practical challenges. However, the approach has fundamental limitations in true exploration due to dependence on expert-generated state distributions, limited task diversity in training, incomplete cost analysis that excludes training overhead, and potential generalization concerns to tasks outside the benchmark distribution.

**Strengths:**

The paper's core strength is developing an autonomous ML agent that learns from experience rather than a prompt-engineered heuristic. The proposed step‑wise RL objective makes online training easier in ML settings with slow experiments. Empirical results show competitive performance of a 7B backbone model compared to Larger 600B size backbone models without training.
The paper systematically validates each component's contribution, showing that all three technical components are necessary. Despite training on only 9 tasks, the model shows meaningful performance gains on 10 held-out tasks.

**Weaknesses:**

1. Different execution times: The paper, although it proposes RL fine-tuning for MLE agents, does not consider a practical problem in training agents specifically for MLE tasks. Different actions from the same state could have different execution times, and if optimized in a vanilla fashion, would lead to more time-consuming yet optimal solutions being explored less. Consider that given a neural network, the agent adds extra layers in one action, opposed to changing the learning rate in a parallel action (people generally perform parallel rollouts for efficiency), then the second type of action would lead to faster rewards and hence be more frequent during training.
2. Reward function is coarse: Mapping all the error cases to -1 and all the valid or corner edits to 0 would treat syntax issues, dependency issues alike, and all the types of out-of-memory issues (model loading OOM, memory leak OOM, and large batch size OOM) the same, although they convey different information and levels of mistakes and understanding. Moreover, the reward considers only the task accuracy and ignores compute, memory, cost, etc., important parameters. Although the reward is simple, it is prone to reward hacking. Another instance of reward hacking could be that the agent learns to find good seeds to make the performance increase, rather than actually learning to improve the architecture or training algorithm.
3. Trajectory Collection is offline with respect to the training policy: The paper states that they use the collected trajectories and at each step, ask the smaller agent to propose an action on a state sampled from the larger agent. This might lead to the problems of distribution shift when using the smaller agent to generate the trajectories from scratch. This might also bias the learning towards expert behavior and limit exploration. This is more akin to a mix of off-policy behavior cloning and on-policy action generation. The agent cannot learn to recover from or explore states outside the expert trajectory distribution, which severely constrains the "learning through experimentation" claim. This raises the question, how would this approach extend to domains where high-quality expert trajectories aren't available?
4. Incomplete Cost Analysis: Figure 2 only compares the inference costs per trajectory but excludes the substantial training and inference costs required for data collection using GPT-4o-mini, supervised fine-tuning, and RL training. For a fair comparison, these costs should be amortized over expected usage.
5. Narrow Task Distribution and Generalization: Training on mainly regression and classification, and then testing on similar tasks raises the question of true cross-task generalization (except one generation task in testing). The paper lacks evidence of generalization to: (a) fundamentally different ML tasks, e.g., if trained on supervised learning, can it handle RL tasks?, (b) different data modalities not seen in training.
6. Evaluation Metric Limitations: The Performance gain delta metric depends heavily on the initial script quality, which may vary across tasks. A task with a poor initial script will show larger gains for the same absolute improvement. The paper would benefit from also reporting: (a) absolute performance metrics, (b) comparison against human expert solutions, and (c) success rate in reaching specific performance thresholds.
7. Script editing using a different model: Note that the training script is not edited by the model being trained. The actual editing is done by a different model (according to Table 4 and prompts shown), yet there is no discussion on which model or agent scaffold was used for this action. Moreover, if the policy being trained is not making edits, why is it penalized for errors induced by the editing model?

**Questions:**

1. How do you prevent the agent from 'reward hacking' by discovering shortcuts, such as finding optimal random seeds, rather than learning generalizable ML engineering improvements?
2. Since the trajectories are collected offline, how does this affect the test time performance of the model when it has to generate an action on its own trajectories that it has never seen during training time? What is the performance of your model when starting from an incorrect or suboptimal state compared to a pre-trained prompt-based method?
3. How do you account for more complex solutions that are less explored during RL due to the high time of execution, leading to fewer occurrences?
4. Which model is used to edit the script?
5. What is the average trajectory length at testing time? What is the average execution time of the solution generated by the trained model?
6. What is the total computational cost, including expert trajectory collection, fine-tuning, and RL training? How many trajectories would need to be run at inference time to amortize this cost compared to a strong prompt-based method directly?
7. How sensitive is the approach to expert trajectory quality? What happens if you use trajectories from a weaker model?
8. Can you provide evidence of generalization to qualitatively different ML tasks? For example, if trained on supervised learning tasks, can it handle reinforcement learning or unsupervised learning tasks?
9. How do you select the baseline script for each task? The reward and evaluation both depend on this selection.

---

> ### Author Response · Authors · 2025-11-27
> **Response to Reviewer i2zT --- Part 1/6**
>
> We thank the reviewer for the constructive comments and suggestions. We address each of your comments in the following.
> > **Q1:** More complex solutions could have more execution times thus lead to fewer occurrences and less exploration.
>
> **A1.**  Thank you for this insightful question. We agree that in many practical agent-training setups, actions with longer wall-clock execution time can be under-explored if rollouts are run in parallel and updates are applied as soon as the first results arrive. However, in our implementation, we design the training pipeline such that longer-running but potentially more powerful solutions are not systematically disadvantaged, for the following reasons:
>
> **Synchronous batch updates:** We collect trajectories in batches and only update the policy after all runs in the batch have finished. Thus, a “slow but strong” solution and a “fast but weak” solution that are sampled in the same batch contribute equally to the update frequency, as long as both finish within a fixed time cap.
>
> **Step-wise RL:** During step-wise RL, we sample states from a pre-collected expert state pool and roll out only a single next action. This decouples policy learning from full training-time cost and allows complex edits (e.g., adding layers) to be explored without being suppressed by their end-to-end runtime.
>
> **Exploration-enriched fine-tuning:** As shown in Figure 7, our exploration-enriched fine-tuning explicitly broadens the action distribution beyond simple hyperparameter tweaks. This ensures that more complex strategies appear frequently enough to be evaluated and explored in step-wise RL.
>
> **Reward module:** The reward is based on normalized performance improvements, not execution speed. Longer-running configurations that yield larger metric gains are rewarded more, while quick changes without improvement are penalized or receive small reward.
>
> Together, these designs ensure that more complex solutions are not systematically under-explored due to longer execution time, and in practice we observe that the trained agent does learn and reuse complex strategies across tasks.

---

> ### Author Response · Authors · 2025-11-27
> **Response to Reviewer i2zT --- Part 2/6**
>
> > **Q2.1: Reward function is coarse.**
>
> **A2.1:** Our reward design is intentionally simple and performance-centric. This is a deliberate modeling choice that matches the scope of this work.
>
> **Why all “hard errors” share the same penalty(-1).** The negative reward is used to enforce a coarse notion of validity: whether the agent’s edit leads to an executable training run or not. Once the agent learns to avoid such invalid states, most of the learning signal comes from metric differences between valid runs. In our experiments, distinguishing between different error subtypes (syntax vs. dependency, etc.) did not bring noticeable benefits, while substantially complicating reward engineering.
>
> **Why all OOM-like “resource corner cases” get a neutral reward(0).** Different OOM events (larger model, larger batch, etc.) can correspond to promising configurations that simply exceed our fixed hardware budget. Since we cannot observe their true performance under sufficient resources, penalizing them as -1 would systematically bias the policy towards overly conservative, small configurations. Assigning 0 keeps them from being rewarded, but also avoids discouraging potentially strong, capacity-increasing ideas.
>
> **Why we focus the reward on task performance instead of explicit compute/memory/cost.**
> Our main goal is to study whether a learning-based agentic ML paradigm can learn generalizable ML engineering behaviors that improve performance on unseen tasks. The benchmark metrics from MLAgentBench and MLE-bench (e.g., accuracy, RMSE, log-loss) are the standard way to measure such improvement and to compare across tasks and methods. In our setting, compute and memory are largely fixed by the benchmark environment (same hardware and time limits), so they act as constraints rather than explicit optimization targets. Incorporating them into the reward would introduce an additional, application-dependent trade-off that is beyond the scope of this work, where we focus on cross-task performance generalization.
>
>
> > **Q2.2: Reward hacking.**
>
> **A2.2:** We are aware of the risk that an agent might exploit superficial shortcuts (e.g., changing random seeds) instead of learning genuine ML improvements. In our current training framework we handle this in three ways:
>
> **Explicit constraints in prompts and environment:** We show the full prompt of task description in Fig. 6. The prompt explicitly forbids changing random seeds. Also, the environment automatically checks for edits to seed-related code, dataset splits, and evaluation functions. Such edits are treated as invalid actions and discarded from RL training.
>
> **Empirical behavior analysis:** We inspected RL trajectories via scripts and did not detect behaviors such as systematically modifying seeds, data splits, or evaluation logic. Instead, the learned edits predominantly focus on architectures, optimizers, regularization, and data augmentation, which is consistent with genuine ML engineering.
>
> **Why this is not a profitable strategy under our RL setup:** Our reward is step-wise and based on metric improvements. Changing seeds without changing the training procedure only induces small, unstable fluctuations around the same expectation, which averages out over many steps and tasks. In contrast, improving architectures or training strategies leads to consistent metric improvements and thus higher expected reward across tasks.
>
> Together, explicit constraints, automatic checks, and the structure of our step-wise differential reward make “seed hacking” both forbidden by design and empirically absent in our experiments.

---

> > ### Author Response · Authors · 2025-11-27
> > **Response to Reviewer i2zT --- Part 3/6**
> >
> > > **Q3.1:** Offline trajectories and distribution shift.
> >
> > Thanks for the question. Our training is not pure offline behavior cloning. Expert trajectories are only used to build a **state pool** and to do a short SFT for **format‑correct and diverse actions**. In the RL stage, for each sampled state we generate a *new* action and assign reward purely from environment execution (our reward module), independent of the expert action. Thus, the policy is optimized to maximize environment reward, not to imitate the expert.
> >
> > Regarding distribution shift: at test time the learned policy rolls out full trajectories and will visit states not exactly seen in training. In practice this does not break performance because (i) SFT is done on diverse trajectories from 9 tasks and an exploration‑enriched action pool, which makes the policy robust and format‑correct across a wide range of situations; (ii) the expert trajectories themselves are noisy and contain failed edits and recovery steps, so the state pool already covers a broad spectrum of suboptimal and erroneous states, not just near‑optimal ones. Empirically, on 10 held‑out tasks, ML‑Agent (7B) achieves on average 16.4% performance gain, consistently outperforming prompt‑based agents driven by much larger models . These results are obtained by rolling out complete trajectories from the initial baseline scripts, indicating that our agent generalizes well to its own trajectory distribution and is not fragile under distribution shift.
> >
> >
> > > **Q3.2:** Sensitivity to expert trajectory quality, weak experts, and suboptimal states.
> >
> > Our method is **not very sensitive to expert trajectory quality**, as long as the trajectories are **format-compatible and diverse enough** to expose the agent to a wide range of actions. **We deliberately keep imperfect trajectories.** We do not filter out expert demonstrations for errors. The subsequent RL stage uses our reward module to distinguish beneficial and harmful actions at each step, so the agent can learn to avoid bad behaviors even if they appear in the expert data.
> >
> > **Role of diversity vs. quality.** Empirically, we observe that what matters is **action diversity**, not that the expert is strong. If expert trajectories collapse to a narrow set of similar edits, RL has little chance to explore alternative strategies. When the trajectories are diverse, even if many actions are suboptimal, the agent can still learn which regions of the action space are promising. This is exactly what our exploration‑enriched fine‑tuning is designed for: it trains the policy to output a broad, format‑correct action distribution that later RL can refine using the reward.
> >
> > **Evidence from a weak expert.** In our experiments the “expert” is already a **relatively weak model (GPT‑4o‑mini)** under the same scaffold. Nevertheless, using this trajectories, our 7B ML‑Agent **consistently and substantially outperforms** this expert across tasks. We report average performance gain over the GPT‑4o‑mini expert on 13 tasks in Table R1.
> >
> >    **Table R1: Average performance gain (\%) compared to the expert GPT‑4o‑mini.**
> >
> >    | Method      | cifar10 | house | feedback | denoising | leaf  | statoil | whale  | learning | detecting | spooky | jigsaw | us     | tabular | Avg.  |
> >    |-------------|---------|-------|----------|-----------|-------|---------|--------|----------|-----------|--------|--------|--------|---------|-------|
> >    | GPT‑4o‑mini | 4.82    | 4.64  | 9.06     | 6.23      | 6.67  | 0.02    | 3.13   | 0.74     | 0.43      | 1.14   | -0.08  | 3.64   | 0.00    | 3.11  |
> >    | ML-Agent    | 33.80   | 6.77  | 13.47    | 52.38     | 13.87 | 1.41    | 72.89  | 1.91     | 1.74      | 1.76   | 0.01   | 12.96  | 0.20    | 16.40 |
> >
> >    Even though GPT‑4o‑mini is far from state‑of‑the‑art and its trajectories contain several mistakes, the learned agent surpasses it by **~13 percentage points on average**, and by a large margin on challenging tasks such as *denoising* and *whale*. This strongly suggests that our approach can leverage weak/noisy expert data and still improve significantly beyond them.
> >
> > **Concerning incorrect or suboptimal states:** because we keep failed and partially‑failed runs, the state pool naturally includes many “bad” states. Step‑wise RL explicitly trains on such states and rewards local improvements, so the agent learns how to **recover from suboptimal states**. In our evaluations, all agents start from the (often suboptimal) initial scripts; yet ML‑Agent achieves larger gains than prompt‑based baselines with much larger backbones on most tasks, indicating that from the same suboptimal starting point, the learned agent is at least as robust—and often more effective—than prompt‑based methods.

---

> > > ### Author Response · Authors · 2025-11-27
> > > **Response to Reviewer i2zT --- Part 4/6**
> > >
> > > >**Q4:** Cost Analysis.
> > >
> > > **A4:** Thank you for raising this important point. We provide a complete cost analysis, including the computational cost of expert trajectory collection, supervised fine-tuning (SFT), and reinforcement learning (RL) training.
> > >
> > > **Table R2:** The computation costs($) before inference.
> > >
> > > |expert trajectory collection | SFT  | RL   | total cost |
> > > |-----------------|------|------|------------|
> > > | 214.47          | 79.20 | 29.14| 322.81     |
> > >
> > > **Table R3:** The number of trajectories need to be run to amortize the cost compared to strong prompt-based methods.
> > >
> > > | Method      | Model             | #Trajectories need to be run to Amortize Cost |
> > > |-------------|-------------------|----------------------------------|
> > > | MLAB        | Deepseek-R1       | 4262|
> > > | MLAB        | Gemini-2.5-pro    |1518|
> > > | MLAB        | GPT-5             | 1633|
> > > | AIDE        | Deepseek-R1       | 3488|
> > > | AIDE        | Gemini-2.5-pro    |2106|
> > > | AIDE        | GPT-5             | 1901|
> > > | ML-Master   | Deepseek-R1       | 3284|
> > > | ML-Master   | GPT-5             | 3945|
> > >
> > > These results show that due to its low per-trajectory inference cost, our ML-Agent requires only a few thousand trajectories to be run to amortize the pre-inference cost compared to strong prompt-based methods.
> > >
> > > > **Q5:** Generalization to qualitatively different ML tasks.
> > >
> > > **A5:** Thank you for raising the concern about narrow task distribution and true cross-task generalization. In the original submission, most evaluations are on supervised regression/classification, plus one image denoising task. To address this more systematically, we additionally evaluate ML-Agent on 5 new held-out tasks that were never used in training, covering unseen modalities and task types. In table R4, we report the average performance gain over 8 trajectories, following our main paper. Across all these ML tasks, ML-Agent consistently improves over the initial script and is competitive with or better than agents based on much larger LLMs (DeepSeek-R1, Gemini-2.5-Pro, GPT-5), providing concrete evidence of cross-task and cross-modality generalization.
> > >
> > > **Table R4:** Cross-task generalization of ML-Agent vs. baselines on qualitatively different held-out ML tasks over 8 trajectories.
> > > | Task Name                                     | Task Type                                           | Metric | Qwen2.5-7B-Instruct | DeepSeek-R1 | Gemini-2.5-Pro | GPT-5       | ML-Agent    |
> > > |-|-|-|-|-|-|-|-|
> > > |Denoising Dirty Documents| image **generation**                                    | RMSE   |  2.10|8.83| 37.85|**66.00**|**52.38**|
> > > | APTOS 2019 Blindness Detection                | **medical image** classification                    | QWK    | -1.10            | -0.61    | -1.51       | -3.35     | **-0.05** |
> > > | H&M Personalized Fashion Recommendations      | **multi-modal (image & text & tabular)** regression | MAP@12 | -4.26             | -1.06     | 2.13        | **20.21** | **8.51**  |
> > > | Optiver – Trading at the Close                | **time series** regression                          | MAE    | 0.02            | 0.038     | 0.06         | **0.14**  | **0.11**  |
> > > | ICML 2013 Whale Challenge – Right Whale Redux | **audio** classification                            | AUC    | 0.11              | 0.18       | **5.88**     | 0.57      | **0.72**  |
> > > | Text Normalization Challenge – English        | text **normalization**                              | Acc.   | -2.49             | 0.08      | **0.13**     | **0.12**  | 0.08      |

---

> ### Author Response · Authors · 2025-11-27
> **Response to Reviewer i2zT --- Part 5/6**
>
> > **Q6:** Evaluation Metric Limitations.
>
> **A6:** We appreciate the reviewer’s concern and clarify our evaluation metric below.
> **Baseline script selection:** For each task, we construct a simple baseline script following the same philosophy as MLAgentBench, i.e., "the starter code mostly implements a simple baseline model that we can compare with during evaluation." Concretely, we use GPT-4o-mini to generate a single initial bug-free training script for each task and fix this script per task. This baseline is shared across all compared agents and all runs, so within a task the “relative performance gain” fairly reflects relative improvements between agents.
> **On the dependence of delta on initial script quality:** We agree that the absolute quality of the initial script  affect the performance gain. Our relative performance gain is used mainly because: (1) it is aligned with MLAgentBench (“average percentage improvement over the baseline in starter code”),  (2) by normalizing improvement relative to the task’s initial performance and metric direction, it also enables more meaningful cross-task comparison despite heterogeneous metrics (accuracy, MAE, log loss, etc.).
> To further address the reviewer’s concerns, we additionally report: 1. **Absolute Performance:** for each task we now provide the absolute score of the baseline script and the absolute scores achieved by each agent (ours and baselines) under the original task metric (e.g., accuracy, MAE, log loss). 2. **Success Rate:** following MLAgentBench, we define success as achieving at least a 10% improvement over the initial script and report the success rate over 8 trajectories) for all agents.
>
> **Table R5:** Absolute performance of ML-Agent vs. baselines over 8 trajectories on 3 held-in and 10 held-out tasks.
> | Task                        | Metric              | Qwen2.5-7B-Instruct | Qwen3-235B | DeepSeek-R1 | GPT-5  | Gemini-2.5-Pro | ML-Agent |
> |--|---------------------|--|------------|-------------|--------|----------------|----------|
> | cifar10                     | Acc.(%) ↑          | 52.19               | 81.14      | 66.39       | 83.12  | 60.12          | 68.88    |
> | house-price| MAE ↓| 21626| 21023      | 20928    | 19043  | 21425          | 20209    |
> | feedback| MCRMSE ↓| 0.6735| 0.6372     | 0.6452      | 0.5960 | 0.6823         | 0.5910   |
> | denoising| RMSE ↓              | 0.1523| 0.0582     | 0.1419      | 0.0529 | 0.0967         | 0.0741   |
> | leaf                        | Log loss ↓          | 0.0780| 0.0817     | 0.0761      | 0.1165 | 0.0835         | 0.0689   |
> | statoil-iceberg             | Log loss ↓          | 0.3092| 0.3385     | 0.2908      | 0.3096 | 0.3033         | 0.2868   |
> | whale                       | MAP@5 ↑             | 0.1304| 0.1472     | 0.1551      | 0.2203 | 0.1422         | 0.2009   |
> | learning (essay scoring)    | QWK ↑               | 0.7422              | 0.7354     | 0.7336      | 0.7652 | 0.7332         | 0.7472   |
> | detecting (insults)         | Acc.(%) ↑          | 80.13               | 80.53      | 79.92       | 88.65  | 79.82          | 81.11    |
> | spooky-author| Log loss ↓          | 0.4515              | 0.4458     | 0.4454      | 0.4189 | 0.4492         | 0.4415   |
> | jigsaw-toxic| AUC ↑| 0.9746| 0.9753     | 0.9752      | 0.9752 | 0.9752         | 0.9753   |
> | us-patent| PCC ↑| 0.4700| 0.4619     | 0.4651      | 0.5589 | 0.4536         | 0.5117   |
> | tabular-playground          | Acc.(%) ↑| 95.94| 95.83| 95.78       | 96.12  | 95.90          | 96.09    |
>
> **Table R6:** Success rate(the percentage over 8 trajectories where the agent achieves an 10%
>  improvement on the performance metric over the initial script) of ML-Agent vs. baselines on 3 held-in and 10 held-out tasks..
>
> | Task| Qwen2.5-7B-Instruct | Qwen3-235B | DeepSeek-R1 | GPT-5  | Gemini-2.5-Pro | ML-Agent (Ours) |
> |--------------------------|---------------------|------------|-------------|--------|----------------|-----------------|
> | cifar10| 0.00| 100.00     | 25.00| 75.00 | 25.00| 100.00|
> | house-price| 0.00| 0.00| 0.00        | 62.50 | 0.00| 12.50|
> | feedback| 12.50| 37.50| 25.00| 50.00 | 0.00| 87.50|
> | denoising| 37.50| 100.00| 12.50| 87.50 | 62.50| 100.00|
> | leaf| 25.00| 0.00| 12.50| 0.00  | 0.00| 62.50|
> | statoil-iceberg| 0.00| 0.00| 0.00| 0.00  | 0.00| 12.50|
> | whale| 0.00| 25.00| 12.50| 100.00| 25.00| 100.00|
> | learning| 0.00| 0.00| 0.00| 0.00  | 0.00| 0.00|
> | detecting| 0.00| 0.00| 0.00| 25.00 | 0.00| 0.00|
> | spooky-author| 0.00| 0.00| 0.00| 25.00 | 0.00| 0.00|
> | jigsaw-toxic| 0.00| 0.00| 0.00| 0.00  | 0.00| 0.00|
> | us-patent| 12.50| 0.00| 0.00| 100.00| 0.00| 37.50|
> | tabular-playground| 0.00| 0.00| 0.00| 0.00  | 0.00| 0.00|
> | Avg Success Rate     | 6.73| 20.19| 6.73| **40.38** | 8.65| **39.42**|
>
> Results show that under these absolute metrics and success‑rate measures, our 7B ML‑Agent remains consistently strong and is competitive with or even better than agents driven by much larger proprietary models.

---

> > ### Author Response · Authors · 2025-11-27
> > **Response to Reviewer i2zT --- Part 6/6**
> >
> > >**Q7:** Script editing using a different model: Note that the training script is not edited by the model being trained. The actual editing is done by a different model (according to Table 4 and prompts shown), yet there is no discussion on which model or agent scaffold was used for this action. Moreover, if the policy being trained is not making edits, why is it penalized for errors induced by the editing model? Which model is used to edit the script?
> >
> > **A7:** In our framework, the **code‑editing component is treated as part of the transition dynamic of the environment**, rather than as the trainable policy. As described in **Appendix B.3**, all script edits during training and evaluating are executed by a fixed code‑editing model:   **Qwen2.5‑Coder‑32B‑Instruct**.   The trainable policy is the ML‑Agent, which operates at the *decision* level: it decides *what kind of edit/action to perform* (e.g., “add batch normalization to the model and increase weight decay”), but it does not directly produce the concrete line‑level code patch. This design choice follows the settings of MLAgentBench**.
> > In addition, we make a standard “reliable editor” assumption: we deliberately use a strong coding model (Qwen2.5‑Coder‑32B‑Instruct) and prompt it to faithfully translate the high‑level action into code and to follow the prescribed action schema. In practice, most compilation/runtime errors arise not from arbitrary editor hallucinations, but from semantically incorrect or underspecified actions issued by the policy (e.g., an action that requests a tensor dimension that does not exist). Therefore, when an error occurs, its root cause is typically the policy’s high‑level action, and not the editing model itself. This is why we assign a negative reward to the policy for such errors:** we want the policy to avoid problematic actions and to produce edit intents that the fixed editor can implement without errors. In other words, the policy is trained to choose high‑level edits that lead to valid and performant code when executed by the MLAgentBench‑style environment (including the code editor). We will clarify this in the revised version.
> >
> > > **Q8:** The average trajectory length at testing time and the average execution time of the solution generated by the trained model.
> >
> > **A8:** Thank you for your question. The **average trajectory length** at test time is 15. The **average execution time** of the solution generated by our trained model for each task is as follows:
> >
> > **Table R7:** The average execution time of the solution generated by ML-Agent.
> >
> > | Task| Average Execution Time (s) |
> > |---|--|
> > | cifar10| 59.289|
> > | house| 14.604|
> > | feedback| 27.591|
> > | denoising| 50.350|
> > | leaf| 6.903|
> > | statoil| 17.264|
> > | whale| 122.379|
> > | learning| 8.877|
> > | detecting| 9.957|
> > | spooky| 13.550|
> > | jigsaw| 43.755|
> > | us| 65.104|
> > | tabular| 101.815|

---

### Official Review · Reviewer_D4Wx · 2025-11-02

**Soundness:** 2
**Presentation:** 3
**Contribution:** 2
**Rating:** 4
**Confidence:** 3

**Summary:**

This paper proposes an online reinforcement learning agent training framework applied to agentic machine learning tasks. The proposed training framework involves three stages: exploration-enriched fine-tuning, step-wise RL, and rewards that are specific to agentic ML tasks. Experimentally, the authors show that with limited training, their 7B agent can match the performance of frontier models such as GPT-5.

**Strengths:**

- This paper provides strong evidence that RL can be used successfully to improve smaller (7B) models to the point of enabling complex agentic behavior on ML-specific tasks, which I view as a strong contribution.
- The proposed technical solutions are quite reasonable. In particular, the authors note that ML agents can suffer from a lack of exploration, and provide an SFT technique that specifically targets this problem. The proposed exploration-enriched fine-tuning technique is both novel and well-executed.
- The results on the tasks that were evaluated suggest that the method works and is competitive with frontier models. This is a strength, however, I have discussed other concerns about the evaluation details in the weaknesses section.

**Weaknesses:**

- The authors train on only 9 tasks from subsets of MLAgentBench and MLE-Bench. How were these tasks chosen? Details about the rationale behind the training task selection seem missing.
- During evaluation, the authors evaluated on a small number of held-out tasks from MLE-Bench. It is unclear how this evaluation subset was selected from the larger set of tasks that were not used during training, even within MLE-Bench. A more convincing evaluation might involve evaluating on entire held-out benchmarks.
- Training requires generating trajectories using GPT-4o-mini (or some other expert model) which might fundamentally limit the complexity of tasks on which it can be used, to tasks that are already solvable by frontier models.

**Questions:**

- Related to the first weakness point, can the authors provide the specific criteria used to select the 9 training tasks?
- MLE-bench contains significantly more than 10 tasks. How were the 10 held-out evaluation tasks selected from the full set of unused tasks?

---

> ### Author Response · Authors · 2025-11-27
> **Response to Reviewer D4Wx --- Part 1/2**
>
> We appreciate the reviewer’s comments. We address each of your comments in the following.
>
> > **Q1:** Training task selection rationale.
>
> **A1:** We thank the reviewer for raising this point. Collecting step-wise expert trajectories for ML agents is computationally expensive because it requires repeatedly running full ML pipelines. Under our compute budget, we therefore prioritized **fast-executable** ML tasks and ensured **diversity**: we selected 9 tasks (5 from MLAgentBench, 4 from MLE-Bench) that jointly cover **4 data modalities**(image, tabular, text, and graph), **2 task types** (classification, regression), and **multiple metrics**(accuracy, AUC, log-loss, MAE, etc.) Thus, the 9 tasks are not arbitrary; they were chosen to balance computational cost and broad coverage of realistic ML settings.
>
> **Empirical evidence.** We explicitly study the impact of the number of RL training tasks (0, 3, 6, 9) in Fig. 5. As the number of tasks increases, performance on both held-in and held-out tasks improves **monotonically**: the average performance gain on held-out tasks rises from ~0% (0 tasks / SFT-only) to ~3%, 6%, and 16% with 3, 6, and 9 tasks, respectively. This indicates that adding more training tasks consistently helps, and our 9-task setup is a **compute-constrained point on a favorable scaling curve**.
>
> **Scalability.** Our method does not rely on any special property of these 9 tasks; with more compute, it can be straightforwardly extended to more tasks. We will clarify this in the revision.
>
>
> > **Q2:** Held-out evaluation task selection.
>
> **A2.** We appreciate the reviewer’s concern and clarify our evaluation setup below.
>
> **Selection criteria.** Among the unused tasks in MLE-Bench (and additional Kaggle tasks), we did not cherry-pick “easy” cases. Under a limited evaluation budget, we selected 10 held-out tasks (none used in training) to provide **representative coverage** of the space of ML pipelines, including:
>
> - modalities: image, text, tabular, time series, audio, and multi-modal (image + text + tabular);
> - task types: classification, regression, generation, normalization;
> - metrics: RMSE, log-loss, MAP@5, quadratic weighted kappa, etc.
>
> Thus, the 10 tasks form a small but representative slice of the unused tasks, aimed at testing generalization to **unseen ML tasks with heterogeneous inputs and metrics**, rather than a single benchmark.
>
> **Additional ML tasks.** To further address this concern, we expand our evaluation with additional held-out tasks from MLE-Bench and the Kaggle platform, covering medical image, audio, time series, and multi-modal settings. The results in Table R1 show that ML-Agent maintains strong cross-task generalization compared with both same-scale open-source baselines (e.g., Qwen2.5-7B-Instruct) and much larger proprietary models:
>
> **Table R1:** Cross-task generalization of ML-Agent vs. baselines on additional held-out ML tasks over 8 trajectories.
> | Task Name | Task Type |Metric| Qwen2.5-7B-Instruct | DeepSeek-R1 | Gemini-2.5-Pro | GPT-5 | ML-Agent |
> |---|---|---|---|---|---|---|---|
> | APTOS 2019 Blindness Detection | medical image classification | QWK|-1.1040 | -0.6148 | -1.5141 | -3.3470 | **-0.0455** |
> | H&M Personalized Fashion Recommendations | multi-modal (image & text & tabular) regression |MAP@12| -4.2553 | -1.0638 | 2.1277 | **20.2128** | **8.5106** |
> | Optiver – Trading at the Close | time series regression |MAE| 0.0229 | 0.0382 | 0.0573 | **0.1412** | **0.1145** |
> | ICML 2013 Whale Challenge – Right Whale Redux | audio classification |AUC| 0.1052 | 0.1804 | **5.8770** | 0.5712 | **0.7215** |
> | Text Normalization Challenge – English | text normalization |Acc.| -2.4940 | 0.0839 | **0.1345** | **0.1243** | 0.0839 |
>
> Across these diverse held-out tasks and metrics, ML-Agent is consistently competitive or superior, especially relative to the same-parameter baselines, supporting our claim that it learns **robust cross-task generalization** rather than overfitting to a small, hand-picked subset. We will clarify the evaluation task selection criteria and include these additional results in the revised manuscript.

---

> > ### Author Response · Authors · 2025-11-27
> > **Response to Reviewer D4Wx --- Part 2/2**
> >
> > > **Q3:** Training requires generating trajectories using GPT-4o-mini (or some other expert model) which might fundamentally limit the complexity of tasks on which it can be used, to tasks that are already solvable by frontier models.”
> >
> >
> > **A3.** Our framework uses GPT‑4o‑mini trajectories only to bootstrap training, not as an upper bound on performance. We rely on expert trajectories to (i) build a diverse, format-correct state pool and (ii) perform exploration‑enriched fine-tuning so that the agent can reliably interact with the environment. In the RL stage, however, the policy is optimized only with respect to our agent-specific reward under the step-wise RL objective, without any imitation loss toward the expert. Thus, the learned policy is free to deviate from and surpass the expert.
> >
> > Empirically, our 7B ML-Agent already outperforms the GPT‑4o‑mini agent by a substantial margin. As shown in Table R2, ML-Agent improves the average performance gain from 3.11% (GPT‑4o‑mini) to 16.40%, and achieves much larger gains on tasks where GPT‑4o‑mini is relatively weak or even fail.
> >
> >
> > **Table R2:** Average performance gain (\%) compared to the expert GPT-4o-mini.
> > | Method         | cifar10 | house | feedback | denoising | leaf  | statoil | whale  | learning | detecting | spooky | jigsaw | us     | tabular | Avg.  |
> > |----------------|---------|-------|----------|-----------|-------|---------|--------|----------|-----------|--------|--------|--------|---------|-------|
> > | GPT‑4o‑mini    | 4.82    | 4.64  | 9.06     | 6.23      | 6.67  | 0.02    | 3.13   | 0.74     | 0.43      | 1.14   | -0.08  | 3.64   | 0.00    | 3.11  |
> > | ML-Agent| 33.80   | 6.77  | 13.47    | 52.38     | 13.87 | 1.41    | 72.89  | 1.91     | 1.74      | 1.76   | 0.01   | 12.96  | 0.20    | 16.40 |
> >
> >
> >
> > More broadly, our method does not assume the expert to be optimal: we only use its trajectories to expose valid regions of the state space. Failures in expert trajectories are penalized by our agentic‑ML reward, and RL actively moves the learned policy away from such behaviors. Thus, expert trajectories act as a starting point for exploration rather than a ceiling on achievable performance.

---

### Meta-Review · Area_Chair_v8Le · 2026-01-06

**Summary:**

The paper proposes upskilling a smaller (cheaper) agent using a combination of RL approaches to match performance of a larger LLM on ML tasks (MLAgentBench, MLE-bench). The reviewers found the motivation meaningful and the approach sensible, but at the same time found the approach relatively standard. They had concerns about the novelty of the approach as well as the strength of the evaluation.

**Reviewer Concerns:**

The authors expressed concerns around the novelty of the RL approach and the evaluation methodology. The authors commented on both concerns, but I think remnants of concerns remain in both cases. The authors were able to allay some concerns such as careful task curation (e.g., across diverse modalities, etc.).

**Reviewer Scores:**

- D4Wx: The reviewer might have increased their score from 4 to 5 or 6.
- i2zT: The authors rebuttal was comprehensive. The reviewer might have increased their score from 4 to 6.
- 6UHw: I think the reviewer would appreciate the response, but might not change their score much (from 2 to 3 maybe).
- ixvU: I think the reviewer would appreciate the response, but might not change their score from 4.

---

### Decision · Program_Chairs · 2026-01-26

Reject